# OPTIMAL TRANSPORT FOR CAUSAL DISCOVERY

**Ruibo Tu**
KTH Royal Institute of Technology
ruibo@kth.se

**Kun Zhang**
Carnegie Mellon University
Mohamed bin Zayed University of Artificial Intelligence
kunz1@cmu.edu

**Hedvig Kjellström**
KTH Royal Institute of Technology
Silo AI
hedvig@kth.se

**Cheng Zhang**
Microsoft Research
Cheng.Zhang@microsoft.com

## ABSTRACT

To determine causal relationships between two variables, approaches based on Functional Causal Models (FCMs) have been proposed by properly restricting model classes; however, the performance is sensitive to the model assumptions, which makes it difficult to use. In this paper, we provide a novel dynamical-system view of FCMs and propose a new framework for identifying causal direction in the bivariate case. We first show the connection between FCMs and optimal transport, and then study optimal transport under the constraints of FCMs. Furthermore, by exploiting the dynamical interpretation of optimal transport under the FCM constraints, we determine the corresponding underlying dynamical process of the static cause-effect pair data. It provides a new dimension for describing static causal discovery tasks while enjoying more freedom for modeling the quantitative causal influences. In particular, we show that Additive Noise Models (ANMs) correspond to volume-preserving pressureless flows. Consequently, based on their velocity field divergence, we introduce a criterion for determining causal direction. With this criterion, we propose a novel optimal transport-based algorithm for ANMs which is robust to the choice of models and extend it to post-nonlinear models. Our method demonstrated state-of-the-art results on both synthetic and causal discovery benchmark datasets.

## 1 INTRODUCTION

Determining causal relationships between two variables is a fundamental and challenging causal discovery task (Janzing et al., 2012). Conventional constraint-based and score-based causal discovery methods identify causal structures only up to Markov equivalent classes (Spirtes et al., 2001), in which some causal relationships are undetermined. To address this challenge, properly constrained functional causal models (FCMs) have been proposed. FCMs represent the effect as a function of its cause and independent noise and can help identify the causal direction between two variables by imposing substantial structural constraints on model classes, such as additive noise models (ANMs) (Shimizu et al., 2006; Hoyer et al., 2008) and post-nonlinear models (PNLs) (Zhang and Hyvärinen, 2009b). While some of the models, such as PNLs, are highly flexible, the constraints are still restrictive and difficult to interpret and relax. Inevitably, the performance of these methods is sensitive to model assumptions and optimization algorithms, especially in real-world applications.

To handle the mentioned issues, we consider FCMs from a dynamical-system view. By augmenting a time dimension for the static causal discovery task, we interpret FCMs with dynamical causal processes under the least action principle (Arnol'd, 2013). The new interpretation connects FCMs with a large class of models in dynamical systems. It then provides more freedom to model causal influences, possibilities to derive new causal discovery criteria, and a potential direction to generalize causal models with identifiable causal direction.

In particular, we exploit the above idea by leveraging the intrinsic connection between FCMs and optimal transport. Optimal transport is originally introduced by Monge (1781), which has been applied in a large range of applications, not only because it is a natural way to describe moving particles (Ambrosio et al., 2012) but also because of its recent improvement in the computational methods (Cuturi, 2013; Kolouri et al., 2019). Recently, it has also been largely applied to generative models for measuring the distance of probability distributions (Arjovsky et al., 2017; Kolouri et al., 2018; Genevay et al., 2018). Among different optimal transport definitions, the $L^2$ Wasserstein distance got extensive applications in statistics (Rachev and Rüschendorf, 1998), functional analysis (Barthe, 1998), et al. (McCann, 1997; Otto, 1997). The dynamical formulation of the $L^2$ Wasserstein distance is introduced by Benamou and Brenier (2000) for relaxing the computational costs. We find that in the context of the dynamical formulation, FCMs can be connected with optimal transport. Furthermore, with the dynamical interpretation of optimal transport, one can naturally understand FCMs from a dynamical-system view, which makes it possible to derive new criteria to identify causal direction. Moreover, it also enables us to develop practical algorithms with optimal transport for static causal discovery tasks without learning a regression model. Our main contributions are:

1. *Dynamical interpretation of FCMs in the bivariate case.* We provide dynamical interpretations of optimal transport under the constraints of FCMs. Furthermore, we introduce a time variable, determine the underlying dynamical process under the least action principle (Arnol'd, 2013) for the static bivariate causal discovery task, and characterize properties of the corresponding dynamical systems (Sec. 3.1 and Sec. 3.2).

2. *A criterion for determining causal relationships between two variables.* We study the corresponding dynamical systems of FCMs and prove that ANMs correspond to volume-preserving pressureless flows. Moreover, based on the divergence of their velocity fields, we propose a criterion for determining causal relationships and show that under the identifiability conditions of ANMs it is a valid criterion for ANMs, which can be extended to PNLs directly (Sec. 3.2).

3. *An optimal transport-based approach (DIVOT) for distinguishing cause from effect between two variables.* DIVOT inherits the advantages of one-dimensional optimal transport. It can be computed efficiently and does not require independence tests, learning a regression model, or deriving likelihood functions for complicated distributions. Experimental results show that our method is robust to the choice of models and has a promising performance compared with the state-of-the-art methods on both synthetic and real cause-effect pair datasets (Sec. 4 and Sec. 6).

## 2 PRELIMINARIES

**Optimal transport: the underdetermined Jacobian problem.** We mainly follow the notations and the definitions of (Benamou and Brenier, 2000). Suppose that two (probability) density functions, $p_0(\mathbf{x})$ and $p_T(\mathbf{x})$ where $\mathbf{x} \in \mathbb{R}^d$, are non-negative and bounded with total mass one. The transfer of $p_0(\mathbf{x})$ to $p_T(\mathbf{x})$ is realized with a smooth one-to-one map $M : \mathbb{R}^d \to \mathbb{R}^d$. *The Jacobian problem* is to find $M$ that satisfies *the Jacobian equation*, $p_0(\mathbf{x}_0) = p_T(M(\mathbf{x}_0))|\det(\nabla M(\mathbf{x}_0))|$, where $\mathbf{x}_T = M(\mathbf{x}_0)$, $\nabla$ is the gradient in vector calculus, and $\det(\cdot)$ denotes determinant. This is an underdetermined problem as many maps can be the solutions. A natural way is to choose the optimal one, e.g., the one with the lowest cost. A common cost function is the $L^p$ Wasserstein distance.

**$L^p$ Wasserstein distance and its one-dimensional closed-form solution.** The $L^p$ Wasserstein distance between $p_0$ and $p_T$, denoted by $W_p(p_0, p_T)$, is defined by $W_p(p_0, p_T)^p = \inf_M \int |M(\mathbf{x}_0) - \mathbf{x}_0|^p p_0(\mathbf{x}_0) d\mathbf{x}_0$, where $p \geq 1$ (Kantorovich, 1948). In this work, we mainly use the square of the $L^2$ Wasserstein distance, denoted by $W_2^2$. Moreover, the one-dimensional (1D) $L^p$ Wasserstein distance has a closed-form solution, e.g., the 1D optimal solution of $W_2^2$ is $M^* = P_T^{-1} \circ P_0$, where $P_0$ and $P_T$ are the cumulative distribution functions for $p_0$ and $p_T$, and "$\circ$" represents the function composition. In practice, the 1D optimal solution can be computed with the average square distance between the sorted samples from $p_0$ and $p_T$ (Kolouri et al., 2019).

**Functional causal models.** FCMs represent the effect $Y$ as a function $f(\cdot)$ of the direct cause $X$ and independent noise $E_y$, where function $f$ describes the causal influence of $X$ on $Y$, and $E_y$ is the exogenous variable/noise. Without any additional assumption on the functional classes, the causal direction is not identifiable (Hyvärinen and Pajunen, 1999; Zhang et al., 2015a). Roughly speaking,

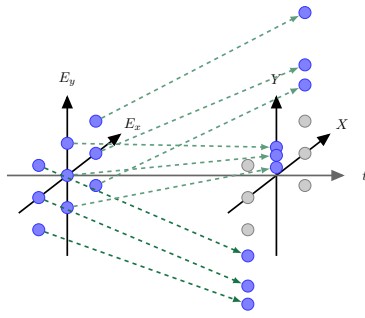
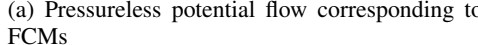

(a) Pressureless potential flow corresponding to FCMs

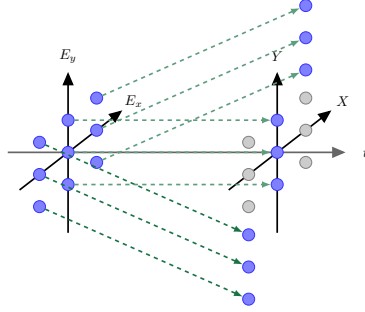

(b) Volume-preserving pressureless potential flow corresponding to ANMs

Figure 1: Panel (a) illustrates the trajectories of the pressureless potential flow corresponding to general FCMs , while panel (b) shows the trajectories of the volume-preserving pressureless potential flow corresponding to ANMs. In panel (b), the trajectories have zero velocities on the $E_x/X$ axis and move along straight lines that are parallel to each other when having the same values of $X/E_x$.

because given variable pair $(X, Y)$, one can always construct $Y = f(X, E_y)$ and another different FCM, $X = \widetilde{f}(Y, E_x)$, such that both of them have independent "noise" (Hyvärinen and Pajunen, 1999; Zhang et al., 2015a). Several works further introduce proper assumptions on model classes, which guarantees that the independence of cause and noise only holds in the causal direction, e.g.,

$$\text{ANM:} \quad Y \;=\; g(X) + E_y; \qquad (1) \qquad \text{PNL:} \quad Y \;=\; h(g(X) + E_y), \qquad (2)$$

where $g$ and $h$ are nonlinear functions and $h$ is invertible.

## 3 DYNAMICAL INTERPRETATION OF FUNCTIONAL CAUSAL MODELS

We first show the connection between FCMs and optimal transport in Sec. 3.1. In Sec. 3.2, we further elaborate the analogy between the optimal transport problem and the causal direction determination problem. We then study the optimal transport under the constraints of FCMs, show the corresponding dynamical systems of FCMs, and characterize the properties of such systems.

### 3.1 A REFORMULATION OF FUNCTIONAL CAUSAL MODELS

As introduced in Sec. 2, FCMs are used to approximate the true data generation process. Given the FCM, $Y = f(X, E_y)$, we rewrite it in the vector form,

$$\mathbf{x}_T = \begin{bmatrix} X \\ Y \end{bmatrix} = \begin{bmatrix} E_x \\ f(X, E_y) \end{bmatrix} = M\Big( \begin{bmatrix} E_x \\ E_y \end{bmatrix} \Big) = M(\mathbf{x}_0), \qquad (3)$$

where $\mathbf{x}_0, \mathbf{x}_T \in \mathbb{R}^2$, their probability densities $p_0, p_T \geq 0$, and $M : \mathbb{R}^2 \to \mathbb{R}^2$. As an analogy to the mass transfer scenario (Monge, 1781), we consider the samples of independent noise $E_x$ and $E_y$ as the particles of materials and regard the map $M$ in Eqn. (3) as a special transformation of the independent noise samples. As shown in Fig. 1, one can consider the data points are transferred from the original positions (which are unmeasured) in the plane $E_x$–$E_y$ at time 0 to the observed positions in the plane $X$–$Y$ at time $T$. Such transformation considers the transfer as a dynamical process which moves the unmeasured independent noise $\mathbf{x}_0 = [E_x, E_y]'$ [1] and consequently leads to the observations $\mathbf{x}_T = [X, Y]'$. From the perspective of FCM-based causal discovery approaches, causal influences are represented by FCMs which represent the effect as a function of its direct cause and an unmeasured noise satisfying the *FCM constraints*:

  (i) *The map constraint:* the values of $X$ are determined by the values of its corresponding noise, i.e., $X = E_x$, while the values of the effect depend on cause $X$ and noise $E_y$;

  (ii) *The independence constraint:* the noise terms are independent, i.e, $E_x$ is independent of $E_y$.

---

[1] "'" denotes the transpose of vectors or matrices.

Note that the optimal transport $M^*$ with the minimal $L^p$ Wasserstein distance is not necessary to be the one in Eqn. (3), because it has no information about the FCM constraints or the true data generation process. In other words, given two sample sets of $\mathbf{x}_0$ and $\mathbf{x}_T$, the couplings given by optimal transport are not necessary to be the ones generated from the ground-truth FCM.

## 3.2 Dynamical interpretation of FCMs: Optimal transport under the FCM constraints

In this section, we jointly consider the causality and the optimality of the maps in the Jacobian problem. It provides both a causal sense of the transformation and a dynamical view of FCMs. We first recap the dynamical formulation of the $L^2$ Wasserstein distance, study such dynamical systems under the FCM constraints, and then show their properties under the FCM and ANM constraints.

**Dynamical $L^2$ Wasserstein distance.** Benamou and Brenier (2000) formulate the $L^2$ Monge-Kantorovich problem as a convex space-time minimization problem in a continuum mechanics framework. Fixing a time interval $[0, T]$, they introduce the concepts of the smooth time-dependent density $\rho(t, \mathbf{x}_t) \geq 0$ and the velocity field $\mathbf{v}(t, \mathbf{x}_t)$. When they are clear from context, we denote them by $\rho$ and $\mathbf{v}$. Because we are considering the bivariate case, $\mathbf{x}_t \in \mathbb{R}^2$ and $\mathbf{v} \in \mathbb{R}^2$. Then, they give the dynamical formulation of $W_2^2$:

$$W_2^2(p_0, p_T) = \inf_{\rho, \mathbf{v}} T \int_{\mathbb{R}^2} \int_0^T \rho(t, \mathbf{x}_t) |\mathbf{v}(t, \mathbf{x}_t)|^2 d\mathbf{x}_t dt, \tag{4}$$

$$\text{s.t.} \quad \begin{cases} \text{initial and final conditions: } \rho(0, \cdot) = p_0, \rho(T, \cdot) = p_T \\ \text{the continuity equation: } \partial_t \rho + \nabla \cdot (\rho \mathbf{v}) = 0 \end{cases},$$

where $\nabla \cdot$ denotes the divergence in vector calculus. They show that minimizing the objective function in the optimization problem (4) is equivalent to finding the dynamical system with the least action (Arnol'd, 2013) and prove that the solutions of (4) are *pressureless potential flows*, of which the fluid particles are not subject to any pressure or force and the trajectories are determined given their initial positions and velocities or given their initial and final positions. Suppose that $M^*$ is the solution given by $W_2^2$. The corresponding flows follow the *time evolution equation*,

$$\mathbf{x}_t = \mathbf{x}_0 + \frac{t}{T} \mathbf{v}(t, \mathbf{x}_t), \text{ where } \mathbf{v}(t, \mathbf{x}_t) = \mathbf{v}(0, \mathbf{x}_0) = M^*(\mathbf{x}_0) - \mathbf{x}_0 \text{ and } t \in [0, T]. \tag{5}$$

The time evolution equation shows that $\mathbf{x}_t$ is a convex combination of $\mathbf{x}_0$ and $M^*(\mathbf{x}_0)$ and that the velocity fields do not depend on time.

As an analogy between the optimal transport problem and causal direction determination, the density $\rho$ and the velocity $\mathbf{v}$ of moving particles can be considered as the probability density and the velocity of changing values of data points. Moreover, the dynamical interpretation of the $L^2$ Wasserstein distance introduces a time variable and provides a natural time interpolation $\rho(t, \mathbf{x}_t)$ of $\rho_0$ and $\rho_T$ together with the velocity field $\mathbf{v}(t, \mathbf{x}_t)$. Similarly, we can also have the natural time interpolation $p(t, \mathbf{x}_t)$ between $p_0$ and $p_T$ as well as the velocity field $\mathbf{v}(t, \mathbf{x}_t)$ under the least action principle (Arnol'd, 2013), which is the dynamical interpretation of FCMs.

**Dynamical $L^2$ Wasserstein distance under the FCMs constraints.** First, we introduce FCM constraints in the context of the dynamical $L^2$ Wasserstein distance. According to the time evolution equation (5), we know that the velocity is fully determined by the initial and final values of $\mathbf{x}_t$. We first consider FCM constraint (i). For the initial and final values of the cause, the value of its observation $X$ is equal to its noise value $E_x$ in Eqn. (3). Consequently, denoting $\mathbf{v} = [v_x, v_y]'$, where $v_x$ and $v_y$ represent the velocities along the $E_x/X$-axis and $E_y/Y$-axis respectively as shown in Fi.g 1, we know that $\forall t \in [0, T]$, $x_t \in \mathbb{R}$, $v_x(t, x_t) = 0$. As for FCM constraint (ii), it implies that the initial noise of cause and effect, corresponding to $E_x$ and $E_y$ at the time 0, are independent. Therefore, we have the FCM constraints for the dynamical $L^2$ Wasserstein distance,

(I) *The map constraints*: $\mathbf{v}(t, \mathbf{x}_t) = [v_x(t, x_t), v_y(t, y_t)]'$, $\forall x_t, y_t \in \mathbb{R}, t \in [0, T]$, $v_x(t, x_t) = 0$;

(II) *The independence constraint*: two random variables (cause and effect) at the initial time have the joint probability density function $p_0(\mathbf{x}_0)$ and they are independent.

Second, we characterize the properties of the dynamical $L^2$ Wasserstein distance under constraints (I) and (II). According to Eqn. (3), the form of $M^*$ is determined as

$$M^*(\mathbf{x}_0) = \begin{bmatrix} E_x \\ f(E_x, E_y) \end{bmatrix};$$ (6)

otherwise, the FCM constraints will be violated. Moreover, the $W_2^2$ under the FCM constraints can be computed with the one-dimensional $W_2^2$ as shown in Prop. 1 (the derivation is in App. D).

**Proposition 1.** *Under constraints* (I) *and* (II)*, the square of the $L^2$ Wasserstein distance between $p_0$ and $p_T$ is*

$$W_2^2(p_0, p_T) = \mathbb{E}_{E_x} \left[ W_2^2 \left( p(E_y), p(Y|E_x) \right) \right].$$ (7)

Furthermore, we consider the constraints of ANMs and characterize the corresponding dynamical systems for now, which can be directly extended to PNLs as mentioned in Sec. 4.2. Based on constraints (I) and (II), we further introduce the *ANM constraint*,

(III) the effect is the sum of noise and a nonlinear function of cause as defined in Eqn. (1).

**Theorem 1** ( *Zero divergence of the velocity field* ). *Under constraints* (I) *and* (II)*, the dynamical systems given by the $L^2$ Wasserstein distance are pressureless flows. Further under ANM constraint* (III)*, they become volume-preserving pressureless flows, of which the divergence of the velocity field,* $\mathbf{v}(t, \mathbf{x}_t) = [v_x(t, x_t), v_y(t, y_t)]'$*, satisfies*

$$div\ \mathbf{v}(t, \mathbf{x}_t) = \frac{\partial v_x(t, x_t)}{\partial x_t} + \frac{\partial v_y(t, y_t)}{\partial y_t} = 0, \forall t \in [0, T],\ x_t, y_t \in \mathbb{R},$$

*where div is the divergence operator in vector calculus.*

Thm. 1 determines the corresponding dynamical systems of FCMs and ANMs by analyzing their densities and velocity fields (the details of the proof are in App. D) and shows an essential property of the corresponding dynamical systems of ANMs. The property indicates a potential criterion for causal direction determination, which the divergence of the velocity field is zero everywhere in the causal direction, while it may not always hold in the reverse direction. Next, we will verify the criterion rigorously, propose an algorithm based on it, and show the extension for the PNL cases.

## 4 CAUSAL DIRECTION DETERMINATION WITH OPTIMAL TRANSPORT

In this section, we define a divergence measure as a criterion for determining causal direction between two variables for ANMs. Based on the criterion, we then provide an algorithm to identify the causal direction, named by the divergence measure with optimal transport (DIVOT).

### 4.1 DIVERGENCE MEASURE AS A CAUSAL DISCOVERY CRITERION

We first define the divergence measure and then show that it is a valid criterion for identifying the causal direction in the bivariate case under the identifiability conditions of ANMs, as shown in Prop. 2 (the proof is in App. D).

**Proposition 2** (Divergence measure as a causal discovery criterion)**.** *Define the divergence measure,*

$$D(\mathbf{v}) = \int_{\mathbb{R}^2} |div\ \mathbf{v}|^2 p_0(\mathbf{x}_0) d\mathbf{x}_0 = \mathbb{E}_{\mathbf{x}_0}[\,|div\ \mathbf{v}|^2],$$ (8)

*where $\mathbf{v} = M^*(\mathbf{x}_0) - \mathbf{x}_0$. Suppose that constraints* (I)*,* (II)*,* (III)*, and the identifiability conditions of ANMs (Hoyer et al., 2008) are satisfied. The divergence measure of the corresponding dynamical system satisfies $D(\mathbf{v}) = 0$ if and only if $X$ is the direct cause of $Y$.*

Nevertheless, there are some challenges to compute the divergence measure. For example, we need to solve a two-dimensional optimal transport problem and compute the derivative of a velocity field for all samples. Another challenge of computing $M^*$ is that we in general have no information about $\mathbf{x}_0$. Such issues are all solved with DIVOT.

## 4.2 PROPOSED METHOD: DIVOT

We first provide an overview of the algorithm, DIVOT, for determining the causal direction between two variables and then introduce the four steps to compute the divergence measure.

**Overview of DIVOT.** Alg. 1 is based on the divergence-measure criterion for causal direction determination and determines causal direction between two variables. Given data of $X$ and $Y$, we want to infer whether $X$ causes $Y$ ($X \rightarrow Y$) or $Y$ causes $X$ ($Y \rightarrow X$). We compute the divergence measure in both directions. In practice, noise can have different variance in different applications, and the larger variance can lead to the larger measure value with finite samples. So we normalize the variance-based measure value with the estimated noise variance. And then the one with the smaller normalized measure value is the causal direction. In App. A, we provide the modified algorithms for including the independent case and the significance of the results with bootstrapping.

---

**Algorithm 1:** DIVOT: divergence measure with optimal transport for causal direction determination.

**Input:** data $\{(x_i, y_i)\}$ and noise distribution $p(E; \theta_0)$
**Output:** $X \rightarrow Y$ or $Y \rightarrow X$.

1 **Def** DIVOT ($\{(x_i, y_i)\}, p(E; \theta_0)$):
2     data, $\theta = \{(x_i, y_i)\}, \theta_0$
3     $\mathcal{L}_{X \rightarrow Y} = \text{Div}(\text{data}, p(E; \theta))$
4     data, $\theta = \{(y_i, x_i)\}, \theta_0$
5     $\mathcal{L}_{Y \rightarrow X} = \text{Div}(\text{data}, p(E; \theta))$

6 **if** $\mathcal{L}_{Y \rightarrow X} < \mathcal{L}_{X \rightarrow Y}$ **then**
7     **return** $Y \rightarrow X$
8 **else**
9     **return** $X \rightarrow Y$

10 **Def** Div (*data*, $p(E; \theta)$):
11     noise = Sampling($p(E; \theta)$)
12     data$^s$, noise$^s$ = OT(data, noise)
13     loss = $\min_\theta$ VarDiv(data$^s$, noise$^s$)
14     **return** loss/Var($p(E; \theta^*)$)

---

**Noise data generation: the first step of computing the divergence measure.** To compute the divergence measure, we need to know the velocity field $\mathbf{v}$ as defined in the time evolution equation (5). It requires the couplings of the data of $\mathbf{x}_0 = [E_x, E_y]'$ and $\mathbf{x}_T = [X, Y]'$. But in the bivariate causal discovery task, only the data of $\mathbf{x}_T$ are given. Therefore, as shown in Line 11 of Alg. 1, we first deal with the issue due to the lack of the noise data of $\mathbf{x}_0$, denoted by $\{(e_x^i, e_y^i)\}$. To obtain the noise data, we may assume a multivariate probability distribution of $\mathbf{x}_0$ with the density $p_0(E_x, E_y)$ and then sample data from it, represented by $(e_x^i, e_y^i) \sim p_0(E_x, E_y)$. Fortunately, due to the FCM constraints, we know that $p_0(E_x, E_y) = p_0(X, E_y) = p(X)p(E_y)$. So we only need to assume the probability distribution of $E_y$ and parameterize it with $\theta$, denoted by $p(E_y; \theta)$. Suppose that the dataset of $\mathbf{x}_T$ with $N$ samples is given, denoted by $\{(x_i, y_i)\}_N$. We first sample a data set of $E_y$ with the sample size $N$, denoted by $\{e_y^i\}_N$, e.g., in the experiments of this work, we use the simplified reparameterization trick,

$$e_y^i = f_\theta^{\text{noise}}(e_y^{\text{source}}) = \theta \times e_y^{\text{source}} \text{ and } e_y^{\text{source}} \sim \mathcal{N}(0, 1)/\mathcal{U}(0, 1), \qquad (9)$$

where $f_\theta^{\text{noise}}$ is a monotonic function, and $e_y^{\text{source}}$ is sampled from a standard normal distribution or a uniform distribution. As the monotonic $f_\theta^{\text{noise}}$ can be very flexible, we can represent flexible noise distributions with monotonic neural networks as in (Huang et al., 2018). Next, we randomly match the data of $X$, denoted by $\{x_i\}_N$, with $\{e_y^i\}_N$, which gives the $\{(x_i, e_y^i)\}_N$ as the data of $\mathbf{x}_0$.

**Optimal transport finds the couplings of the observation and the generated noise.** Computing the divergence measure requires the couplings of the data of $\mathbf{x}_0$ and $\mathbf{x}_T$ because of $\mathbf{v} = M^*(\mathbf{x}_0) - \mathbf{x}_0$; in other words, given the data of $\mathbf{x}_0$ and $\mathbf{x}_T$, we need to solve a two-dimensional optimal transport problem, which gives $M^*$ and the couplings as in Line 12 of Alg. 1. According to Prop. 1, solving the two-dimensional optimal transport problem under the FCM constraints is equivalent to solving one-dimensional optimal transport problems. More specifically, the expectation in Eqn. (7) can be computed with the Monte Carlo estimator by using $N$ samples of $E_x$, denoted by $\{e_x^i\}_N$, and then the two-dimensional $W_2^2$ in Eqn. (7) is computed with $N$ one-dimensional $W_2^2$, i.e., $W_2^2(p_0, p_T) \approx \frac{1}{N} \sum_i W_2^2(p(E_y), p(Y|E_x = e_x^i))$. Importantly, solving the one-dimensional optimal transport problem or computing the one-dimensional $W_2^2$ can be implemented with a sorting operation as in (Kolouri et al., 2019). Therefore, the couplings are found by which given each value of $E_x$, we find the corresponding values of $E_y$ and $Y$ from $\{(e_x^i, e_y^i)\}_N$ and $\{(x_i, y_i)\}_N$ and then match such sorted values of $E_y$ and $Y$.

**Variance-based divergence measure.** Suppose that the previous step has found the couplings, i.e., given any sample $(e_x^i, e_y^i)$, we have its corresponding $(x_i, y_i)$. Then, we will compute the divergence of a velocity field as in Line 13 of Alg. 1. According to the definition, $|\text{div } \mathbf{v}|^2 = (\frac{dv_y}{dy})^2$. Since all the $e_y^i$ are matched with $y_i$, we have $v_y^i = y_i - e_y^i$. A straightforward way to approximate the derivative is using its nearest neighbour pair $(e_{xb}^i, e_{yb}^i)$ and $(xb_i, yb_i)$ and then approximate it with $\frac{d}{dy}(v_y)|_{y=y_i} = \frac{(y_i - e_y^i) - (yb_i - e_{yb}^i)}{y_i - yb_i}$; however, it suffers the following issues (especially in the few-sample scenario) : (a) the denominator is in general a small number, and the distance to the nearest neighbour can be large in the few-sample case, which makes the computation unstable and inaccurate; (b) the deviation on the $X$-axis makes the approximation a biased estimate especially when the gradient of $g(X) + E_y$ at $X = x$ is large. Therefore, we propose the variance-based divergence measure working better in practice. It is straightforward to see that under constraints (I) and (II), the value of the divergence measure of an ANM is zero if and only if the value of the *variance-based divergence measure* is zero, which is defined as

$$D_{\text{var}}(\mathbf{v}) = \mathbb{E}_X[\mathbb{V}[V_{y|X}|X]], \tag{10}$$

where $\mathbb{V}[V_{y|X}|X]$ represents a conditional variance of the velocity field $V_y$ (a random variable) at position $X$ at the initial time. For example, $\mathbb{V}[V_{y|x}|X = x]$ represents the variance of all the velocities $v_y$ at the positions where $X = x$ at the initial time. $\{(x_i, y_i)\}_N$ denotes the cause-effect pair dataset with the sample size $N$. $\{v_{y|x}^i\}_{N_x}$ denotes all the velocities $v_y^i$ at position $X = x$ at the initial time, where the sample size is $N_x$, and their mean value is $\overline{v}_{y|x}$. Then, $\mathbb{V}[Vy|x|X = x] = \sum_{i=1}^{N_x}(v_{y|x}^i - \overline{v}_{y|x})^2/(N_x - 1)$, and

$$D_{\text{var}}(\mathbf{v}) \approx \frac{1}{N} \sum_{x \in \{x_i\}_N} \frac{\left\|\text{sort}(\overrightarrow{y_x}) - \text{sort}(\overrightarrow{e_y}) - \text{ave}(\overrightarrow{y_x} - \overrightarrow{e_y})\right\|_2^2}{N_x - 1}, \tag{11}$$

where $\{x_i\}_N$ represents the set of all the values of $X$ and its sample size is $N$; $\overrightarrow{y_x}$ is the vector of the $Y$ samples where $X = x$; $\overrightarrow{e_y}$ is the vector of the $E_y$ samples where $E_x = x$; $\text{sort}(\cdot)$ sorts a vector; $\text{ave}(\cdot)$ computes the vector mean; and $\|\cdot\|_2^2$ is the square of a $\ell_2$ norm.

**Minimization w.r.t $\theta$.** Given the data $\{(x_i, y_i)\}_N$ and the density $p(E_y; \theta)$, we can compute the divergence measure with the variance-based method. Note that we only initialize $\theta$ with some random value, and $p(E_y; \theta)$ is not necessary to be the true distribution or even significantly different from the true one, which can lead to the wrong result of the divergence measure. Therefore, as shown in Line 13 of Alg. 1, we minimize the divergence measure w.r.t. $\theta$. For the minimization, one can derive the gradient w.r.t $\theta$ in a simple parameterization case as (9), while in the complex case one can use auto-differentiation. According to Prop. 2, the divergence measure in the causal direction is zero if and only if $p(E_y; \theta^*)$ with the optimal parameter $\theta^*$ is the true noise distribution a.e., implied by the identifiability of ANMs. In this work, we used `autograd` and `RMSProp` (gradient descent) of JAX for the minimization.

**Extension to PNLs.** The measure (11) can be directly extended to the PNL cases. Since $h$ in PNL (2) is an invertible function, by considering $h^{-1}(Y)$ as a new random variable, $h^{-1}(Y) = g(X) + E_y$ is an ANM. Thus, for PNLs, under the identifiabililty conditions of PNLs (Zhang and Hyvärinen, 2009b), Prop. 2 still holds; and we only need replace $\overrightarrow{y_x}$ in (11) with $\overrightarrow{\tilde{y}_x} = f_\omega^{\text{PNL}}(\overrightarrow{y_x})$, where $f_\omega^{\text{PNL}}$ is an invertible function, e.g., it can be the simplified version of (Zhang et al., 2015a),

$$\overrightarrow{\tilde{y}_x} = f_\omega^{\text{PNL}}(\overrightarrow{y_x}) = \overrightarrow{y_x} + \omega_a \cdot \tanh(\omega_b \cdot \overrightarrow{y_x} + \omega_c), \tag{12}$$

where $\omega_a$, $\omega_b$, and $\omega_c$ are positive scalars. Moreover, $f_\omega^{\text{PNL}}$ can also be the monotonic neural networks, such as (Huang et al., 2018).

**Extension to the multivariate case.** For simplicity and clarity of the paper, we focus on elaborating the connection between FCMs and dynamical systems and developing the theoretical basis and the method in the bivariate case, which are essential for the further development of FCM-based causal discovery methods. In the case of multiple variables, one can use a constraint-based method to find the causal skeleton (the undirected causal graph) and then use the extension of our method for the edge orientation similar as (Zhang and Hyvärinen, 2009b; Monti et al., 2020; Khemakhem et al., 2021). See App. B for details.

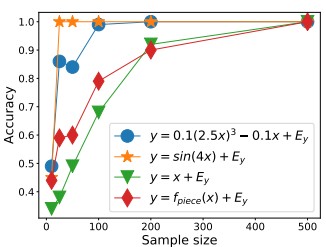

(a) Accuracy on different ANM datasets.

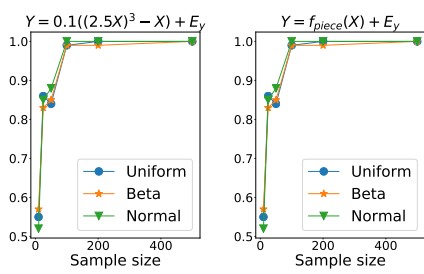

(b) Robustness to prior misspecification.

Figure 2: Performance on synthetic data generated from different ANMs with different sample sizes. $f_{\text{piece}}$ is a discontinuous function: $f_{\text{piece}}(x) = 0.5x^3 - x, x \leq 0$; $f_{\text{piece}}(x) = 1 - 0.5x^3 + x, x > 0$. In panel (a), each result represents the percentage of correct results on datasets with a sample size generated from an ANM. Panel (b) shows the results of applying DIVOT using different types of distributions, $p(E_y; \theta)$, to the datasets with uniform distribution noise $E_y \sim \mathcal{U}(0, 1)$.

## 5 RELATED WORK

There are mainly two types of causal discovery methods for static causal direction determination between two variables. The first one introduces model assumptions to achieve the identifiability of causal direction. Most of such methods are based on LiNGAM (Shimizu et al., 2006) and ANMs (Hoyer et al., 2008; Mooij et al., 2009). Some of them are based on the more general models, e.g., PNLs using MLP for representing nonlinear functions (PNL-MLP) (Zhang and Hyvärinen, 2009b) and PNLs using warped Gaussian process and mixture of Gaussian noise (PNL-WGP-MoG) (Zhang et al., 2015a). Recently, Khemakhem et al. (2021) propose an autoregressive flow-based model (CAREFL), of which the assumption is more general than ANMs and stricter than PNLs. They commonly apply (non)linear regression to learn the function $g$ in (1) (or together with $h$ in (2)), and then test the independence between the independent noise $E_y$ in (1) (the residual) and the cause $X$. And the independence test is commonly Hilbert-Schmidt independence criterion (HSIC) (Gretton et al., 2005); however, as argued by Yamada and Sugiyama (2010), the kernel width limits its practical use and its common heuristic value limits the flexibility of the function approximation. Moreover, there are other criteria proposed in the first type of methods, such as the likelihood ratio (LLR) (Hyvärinen and Smith, 2013), maximum likelihood-based criterion (MML), regression error-based causal inference (RECI) (Blöbaum et al., 2018), and mutual information (Zhang and Hyvärinen, 2009a; Yamada and Sugiyama, 2010). Nevertheless, the common issue of the first type of methods is that they restrict the model classes and that their performance is sensitive to model assumptions. The second type of methods achieves identifiability by proposing other principles instead of restricting model classes, such as exogeneity (Zhang et al., 2015b), the randomness of exogeneous variables (Entropic) (Compton et al., 2021), and independent causal mechanisms, e.g., GPI (Stegle et al., 2010) and IGCI (Janzing et al., 2012).

## 6 EXPERIMENTS

We demonstrate and evaluate our method on the synthetic and real-world cause-effect pair data (Mooij et al., 2016). Moreover, we also provide the experiments and the discussion of our method in the presence of unknown confounding in App. C. The details about experiments are in App. F.

**Synthetic data.** We evaluate DIVOT with Gaussian noise as in (9) on the datasets with different (non)linear functions and samples sizes. We generate synthetic data with the ANMs: 1) $Y = X + E_y$; 2) $Y = 0.1(2.5X)^3 - 0.1X + E_y$; 3) $Y = \sin(4X) + E_y$; 4) if $x < 0, Y = 0.5X^3 - X + E_y$ and if $x \geq 0, Y = 1 - 0.5X^3 + X + E_y$, where $X \sim \mathcal{U}(-1, 1)$ and $E_y \sim \mathcal{U}(0, 1)$ are uniform distribution. For each ANM, we generate datasets with the sample sizes 10, 25, 50, 100, 200, and 500. For each sample size, 100 different datasets are generated. Fig. 2a shows that DIVOT consistently recovers the causal direction for all the cases. Since DIVOT has no smoothness constraint of functions, it can deal with the case 4) which is a discontinuous function. We compared DIVOT with the results of CAREFL (Khemakhem et al., 2021), RECI (Blöbaum et al., 2018), and

Table 1: Percentage (%) of recovering the true causal direction on the Tübingen cause-effect pair datasets (Mooij et al., 2016) compared with the reported results of (Stegle et al., 2010) in the upper table and (Zhang et al., 2015a) in the lower table.

| Ours (PNL) | Ours (ANM) | LiNGAM | ANM-Gauss | ANM-MML | ANM-HSIC | PNL | GPI-HSIC | GPI-MML | IGCI |
|---|---|---|---|---|---|---|---|---|---|
| **78.9±3.9** | 71±4 | 62±3 | 45±3 | 68±1 | 68±3 | 67±4 | 62±4 | 72±2 | 76±1 |

| Ours (PNL) | Ours (ANM) | ANM | WGP-Gauss | WGP-MoG | PNL-MLP | GPI | IGCI |
|---|---|---|---|---|---|---|---|
| **76.6±4.5** | 67±3 | 63 | 67 | 73 | 70 | 72 | 73 |

other benchmark methods in Appendix. As shown in Fig. 7 of App. F, our method performs better than the others. Moreover, we also show the robustness of DIVOT to the prior misspecification as the experiments in CAREFL (Khemakhem et al., 2021). The synthetic data are generated with uniform distribution noise $E_y \sim \mathcal{U}(0, 1)$, where the $p(E_y; \theta)$ of DIVOT is either uniform distribution $\mathcal{U}(0, 1)$, beta distribution $\mathcal{B}(a = 0.5, b = 0.5)$, or standard normal distribution $\mathcal{N}(0, 1)$. As shown in Fig. 2b, DIVOT with the misspecified noise distributions has similar performance with the one using the correct class of distributions.

**Tübingen cause-effect pair dataset.** We apply DIVOT to the Tübingen cause-effect pair dataset (Mooij et al., 2016). This is a collection of real-world cause-effect pairs. We use the variance-based divergence measure (11) and parameterize $p(E_y; \theta)$ as in (9). As for the PNL extension, we implement (12) without the positivity constraint (in practice imposing no constraint on $\omega$ also performs well in this simple formulation). Moreover, we found that given a value of $X$ or $Y$ of the datasets, there are often few samples. Thus, we consider a range of values of $X$, which may introduce bias of the divergence measure; hence we also use a linear debiasing function for reducing the bias. Then, we minimize the variance-based divergence measure over parameters with `autograd` of JAX. More details about the debiasing function and optimization can be found in App. F. To compare with the results reported in other works, we use the maximum number of the sample size $N = 500$ as (Stegle et al., 2010; Zhang et al., 2015a) and run all the experiments with 3 random seeds as (Stegle et al., 2010), though DIVOT is efficient enough for datasets with larger sample sizes as shown in Appendix. In addition, we normalize data and select the ones within 2 standard deviation, because DIVOT is sensitive to the outliers as optimal transport. As shown in Tab. 1, our proposed DIVOT, especially the extension for PNLs, outperforms than the other methods. The reported results in Tab. 1 are taken from (Stegle et al., 2010) and (Zhang et al., 2015a). And our results are based on the same datasets as them, i.e., with 68 and 77 cause-effect pairs respectively. Moreover, the entropic causal inference (Compton et al., 2021) is reported with 64.21% accuracy; CAREFL (Khemakhem et al., 2021) is reported with 73% accuracy on 108 pairs; RECI (Blöbaum et al., 2018) is reported with 76% weighted accuracy on the 100 pairs. From Tab. 1, we can also see that the other ANM/PNL-based methods are more sensitive to the choice of noise distributions. As shown in Tab. 1, the ANM/PNL with Gaussian noise performs worse than the one with a more complex distribution (mixture of Gaussian) or the one combined with a more stable measure, such as HSIC independence test. In contrast, the PNL extension of DIVOT with the Gaussian noise has a state-of-the-art result.

## 7 CONCLUSION

In this paper, we provide a new dynamical-system perspective of FCMs in the context of identifying causal relationships in the bivariate case. We first demonstrate the connection between FCMs and optimal transport and study the dynamical systems of the optimal transport under constraints of FCMs. We then show that FCMs correspond to pressureless potential flows and that ANMs as a special case corresponding to the pressureless potential flows with the velocity field divergence equal to zero. Based on such findings, we propose a divergence measure-based criterion for causal discovery and provide an efficient optimal transport-based algorithm, DIVOT, for identifying causal direction between two variables. The experimental results on both synthetic and real datasets show that compared with the state-of-the-art methods, DIVOT has state-of-the-art results and promising properties for flexibility, efficiency, and prior misspecification robustness. We hope that the connection between FCMs and optimal transport has the potential of helping understand general FCMs from the dynamical perspective and inspiring more generic causal discovery methods.

## ACKNOWLEDGEMENTS

KZ would like to acknowledge the support by the National Institutes of Health (NIH) under Contract R01HL159805, by the NSF-Convergence Accelerator Track-D award ♯2134901, and by the United States Air Force under Contract No. FA8650-17-C7715. RT would like to acknowledge the funding support of the Swedish e-Science Research Centre.

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

APPENDIX

In App. A, we provide the modification of Alg. 1 for including the independent case and the significance of the results. In App. B, we provide the outlook of the extension to the multivariate case. In App. C, we provide the analysis, the discussion, and the experimental results of our method in the presence of unknown confounding. In App. D, we provide the proofs of Prop. 1, the extension of Prop. 1, Prop. 2, and Thm. 1. In App. E, we include the identifiability conditions of ANMs in (Hoyer et al., 2008). In App. F, we introduce the details of the DIVOT implementation and the experiments:

- App. F.1: introduce the concept, position, for computing the divergence measure (11);
- App. F.2: show a potential problem of computing the variance-based divergence measure in the finite/limited-sample case and introduce batches of data to deal with the problem;
- App. F.3: show that using batches can introduce bias and lead to wrong causal relationships (especially in the few-sample case) and then introduce debiasing functions to reduce the bias, which is used for the experiments on real-world data in Sec. 6;
- App. F.4: introduce the optimization methods used in the experiments on synthetic and real-world data in Sec. 6 and show the convexity of the objective function, the divergence measure (11), under the construction in Sec. 6;
- App. F.5: show the robustness of DIVOT to prior misspecification;
- App. F.6: compare DIVOT with other benchmark methods;
- App. F.7: show the efficiency of DIVOT with its running time.

## A    MODIFICATION OF ALG. 1 FOR INCLUDING THE INDEPENDENT CASE AND THE SIGNIFICANCE OF THE RESULTS

The proposed measure is able to deal with the independent case without relying on other methods or tests after adapting the output conditions of the algorithm. The values of our proposed measure in the independent case are zero in the two directions, while the measure values of the causal case are a zero value in one direction and a non-zero value in another direction. The modified output conditions are shown in Tab. 2. To determine that the value of our measure is zero or not, one could follow a

Table 2: Modified output conditions for the independent case.

| Output of DIVOT | $\text{Div}(X \to Y) = 0$ | $\text{Div}(X \leftarrow Y) = 0$ |
|---|---|---|
| X, Y independent | True | True |
| X→Y | True | False |
| X←Y | False | True |

similar way as (Blöbaum et al., 2018) choosing a threshold for real-world applications practically. However, different applications may have different thresholds. For example, in practice, when the noise distributions have different variance in the finite/few sample scenario, the larger variance can lead to the larger measure value. Although one can handle the problem well by normalizing the variance-based measure value with the estimated noise variance, deriving a statistical test for the finite/few sample case is still the ideal way, which is another nontrivial task without assuming the type of noise distributions and will be the future work of our method.

**Bootstrapping for the significance of the results.**    Instead of testing whether a measure value is significantly zero or non-zero, we suggest using a bootstrapping method and testing whether the two measure values are significantly different (one could also test whether the difference between the two measure values are significantly zero or not):

Step 1. use bootstrapping (resampling with replacement) to get B (e.g., 50) bootstrapping datasets;
Step 2. compute two measure values in the two directions for each bootstrapping dataset;
Step 3. apply a two-sample test (T-test) to check whether the mean of one measure value in one direction is significantly different from another one in the other direction.
Step 4. if two measure values are significantly different, we conclude that the smaller one is in the causal direction; if they are not significantly different, we conclude that it is the independent case.

Moreover, we did experiments for the case where the causal relationship is so weak that the FCM with a causal relationship is similar to the independent case (e.g., when the coefficient of the direct cause is extremely small in the linear case, it is close to the independent case where the measure values are close). We generate four datasets with 1000 samples for the four FCMs in the synthetic data experiments in Sec. 6 with a weighting factor $w$: $Y = w \times f(X) + E_y$, and then use the bootstrapping method and T-test. Tab. 3 shows the p-values, of which small values indicate the significantly different measure values. If the significant level is 0.05, our method can tell the difference between two measures in the causal case when $w > 0.02$.

Table 3: Experimental results of testing the significance of the results.

| w | 0.01 | 0.02 | 0.03 | 0.04 | 0.05 |
|---|---|---|---|---|---|
| M1 | 0.155 | 0.038 | 0.0006 | 2.429e-08 | 3.091e-11 |
| M2 | 0.251 | 0.070 | 0.0003 | 0.0001 | 1.935e-07 |
| M3 | 0.390 | 0.119 | 0.0196 | 0.0003 | 9.578e-08 |
| M4 | 0.408 | 0.098 | 0.0037 | 2.571e-05 | 2.346e-11 |
| w | 0.01 | 0.02 | 0.03 | 0.04 | 0.05 |
| M1 | indep | X cause | X cause | X cause | X cause |
| M2 | indep | indep | X cause | X cause | X cause |
| M3 | indep | indep | X cause | X cause | X cause |
| M4 | indep | indep | X cause | X cause | X cause |

## B    EXTENSION TO THE MULTIVARIATE CASE

**The problem of the naive solution to the multivariate case.**    A direct extension to the multivariate case is as the extensions of other bivariate causal discovery methods, such as (Khemakhem et al., 2021), (Zhang and Hyvärinen, 2009b), and (Monti et al., 2020). One can first apply constraint-/score-based methods to get a causal skeleton, an undirected causal graph, and then use the extension of our measure for finding all the causal directions. A problem of the extension in some of the other bivariate works assuming causal sufficiency is that they directly applied their bivariate methods to each edge of the causal skeleton without considering the DAG structure. This can lead to the wrong results, especially in the case where they disregard the common parent/confounder of an edge.

**Extending our method directly to the multivariate case.**    We provided the extensions of Prop. 1 and the variance-based measure, which can be used for computing the measure value considering the DAG structure and then orienting the edges based on the causal skeleton in the multivariate case. Our extension to the multivariate case has the following properties: Given the causal skeleton,

1. it distinguishes Markov equivalent classes under the identifiability conditions of ANMs (or PNLs);

2. it still benefits from the closed-form 1D optimal transport solution and doesn't have the computational issue as in the high-dimensional optimal transport methods because of the FCM constraints;

3. it provides a score of which the causal structure has the minimum value compared with all DAGs of the causal skeleton; moreover, the measure value of a DAG is the summation of all measure values of the causal modules/conditionals/mechanisms.

First, we show the derivation of the extension of Prop. 1 for ANMs, given which the derivation in the PNL case is straightforward (one can consider what we did for Eqn. (12) in our paper). The proposition shows how to efficiently compute the $L^2$ Wasserstein distance between high dimensional distributions under FCM constraints. Suppose that the general ANM is $X_i = g_i(PA_i) + E_i$, where $i = 1, ..., m$, $E_i$ is the noise term of $X_i$, and $PA_i$ denotes the parent variables of $X_i$. The square of $L^2$ Wasserstein distance is

$$W_2^2(p_0, p_T) = \sum_{i=1}^{m} \mathbb{E}_{PA_i}[W_2^2(p(E_i), p(X_i|PA_i)],$$

of which the derivation is shown in App. D.2.

Second, as a direct implication of Thm. 1, given the couplings of $E_i$ and $X_i$, the corresponding dynamical system has zero divergence of its velocity field; in other words, the corresponding dynamical system which moves the samples of $\mathbf{x}_0$ to the samples of $\mathbf{x}_T$ under ANM constraints has zero divergence on each dimension. And the variance-based measure is

$$D_{\mathrm{var}}(\mathbf{v}) \approx \sum_{i=1}^{m} \frac{1}{N} \sum_{k \in \{\text{samples of } PA_i\}_N} \frac{||\mathrm{sort}(\overrightarrow{x_{i|PA_i=k}}) - \mathrm{sort}(\overrightarrow{e_i}) - \mathrm{ave}(\overrightarrow{x_{i|PA_i=k}} - \overrightarrow{e_i})||_2^2}{N_k - 1},$$

where $\overrightarrow{x_{i|PA_i=k}}$ is the data vector, of which the elements are the $X_i$ values of the samples with $PA_i$ taking the value $k$; $\overrightarrow{e_i}$ is the data vector, of which the elements are the generated noise samples; $N_k$ represents the length of the vector $\overrightarrow{x_{i|PA_i=k}}$ or $\overrightarrow{e_i}$.

Next, as the direct extension, one could enumerate all possible DAGs of the causal skeleton and compute their measure values, of which the minimum value is corresponding to the causal graph. Because the causal skeleton is given, it must be the case where one of the two variables of an edge is the cause and the other one is the effect. So the enumerated graphs have two situations: 1) all the edges are correctly oriented; 2) the causal direction of at least one edge is wrong such that the measure value of at least one causal module is significantly larger than the correct one (note that considering a child as the direct cause leads to increasing the measure value, while omitting a cause is not necessary to increase the measure value of the causal module). Therefore, we can simply choose the graph with the minimum measure value as the causal one. As mentioned in the paper, it is also very important to develop practical algorithms for large-scale real-world problems, and there are some points for future works to further explore:
1. Testing the significance of the results in the multivariate case, i.e., whether the minimum one is significantly smaller than the others. One could apply bootstrapping to the dataset and have a p-value for the measure value. One could also apply bootstrapping for each causal module, however, there may exist the problem of the multiple statistical test issue with family-wise errors in this way.
2. Developing an efficient search algorithm in the multivariate case without relying on constraint-/score-based methods. This requires to analysing the measure value of a causal module in more situations (e.g., omitting a parent, involving an independent variable/non-child descendant/non-parent ancestor, or a case mixing the mentioned factors) and considering the characteristics to develop an efficient search algorithm similar as the greedy search algorithm (Chickering, 2002).

## C    UNKNOWN CONFOUNDING

The unknown confounding has different influences on the results of our method in different situations: 1) *independent case:* when two variables are independent; 2) *causal case:* when there is a causal relationship between two variables. Depending on how the unknown confounding influences the pair of variables, it can make our method

1. reverse the direction of the result in the causal case;

2. disregard the causal relationship in the causal case;

3. introduce extraneous causal relationship in the independent case.

To understand the results, we could analyze some toy examples intuitively. One extreme *independent case* is that $X = U$ and $Y = E_y + U$, where $U$ is the unobserved confounder; $E_y$ is the noise of $Y$; $X$ has no noise. Then our method will show that $X$ is the cause of $Y$, even though there is no causal relationship. Similarly, suppose that $U$ and the noise of $X$ and $Y$ have the same distribution with variance equal to 1, and that $X = E_x + 1000U$ and $Y = E_y + U$. Then, the distribution of $X$ can be dominated by $U$, which leads to the wrong result for the same reason as the extreme case.

In the *causal case*, it follows the same reason. For example, $X = E_x + U$ and $Y = X + E_y + 1000U$. Then the distribution of $Y$ can be dominated by $U$. Therefore, the FCM is close to $X = E_x + U$ and $Y = 1000U$, which leads to reversing the causal direction in the result. Moreover, when $X = E_x + 1000U$ and $Y = X + E_y + 1000U$, then both distributions of $X$ and $Y$ can be dominated

by $U$; hence, it is close to $X = 1000U$ and $Y = 1000U$, which leads to disregarding the causal relationship in the result.

Nevertheless, in practice, the impact of unknown confounding is more complex and all factors can be mixed together with the impact of finite samples and the function properties. Therefore, we provide the experiments based on the synthetic datasets. From the experimental results, we can also find that the strength of the confounding and the difference of the confounding strength on the two variables are two important factors resulting in the wrong results of our method.

**Experimental results for the unknown confounder.** We generate a dataset with 1000 samples for each FCM:
FCM1) $X = U$ and $Y = U$;
FCM2) $X = E_x + w_x \times U$ and $Y = Ey + w_y \times U$;
FCM3) $X = E_x + w_x \times U$ and $Y = f(X) + E_y + w_y \times U$,
where $E_x$ and $E_y$ are the noise terms; $U$ is the unknown confounder; $w_x$ and $w_y$ are the coefficients of $U$ representing the confounding strength; we used $f(X) = X$ and $f(X) = \sin(4X)$ in the experiments. As for the data generation, $E_x$, $E_y$, and $U$ follow the uniform distribution, $\mathcal{U}(0, 1)$, and we vary the coefficients $w_x$ and $w_y$.

Instead of testing whether a result is significantly close to zero, we use bootstrapping to resample 50 datasets for each generated dataset and then apply a two-sample test to test whether the measure values in the two different directions are significantly different. Because although the measure value, in theory, is zero in the causal direction when there is no unknown confounder, with finite samples the measure value will be larger than zero in practice. And to decide which measure value is close to zero or not, it requires a threshold which can vary in different applications, or deriving a statistical test which is nontrivial for the measure without assuming the type of noise distributions and can be the future work of our method. Thus, we test whether two measure values are significantly different, and if so, we then conclude that the smaller one is in the causal direction; if they are not significantly different, we then conclude that it is the independent case. As for the experimental results, when the p-value is close to zero, it means that the two measure values are significantly different. And in general, one could take the significant level at 0.05 to make a conclusion. We used 0.05 for the experiments.

The experimental result of FCM1 is that the p-value is equal to 1.0, which means that the measure values in the two directions are not significantly different.

As for the experimental results of FCM2, we found that in the independent case, when the coefficients $w_x$ and $w_y$ are the same, we can get the correct results; when the coefficients are different, the method will give the wrong results. In different applications/scenarios, there are different tolerance ranges for our methods such that when the difference of the confounding coefficients is within the range, even if the confounding coefficients are different, we can still get the correct results.

Table 4: The experimental results of FCM2 (the independent case) in the presence of unknown confounding.

| $w_x \backslash w_y$ | 0.1 | 1.0 | 10 | | $w_x \backslash w_y$ | 0.1 | 1.0 | 10 |
|---|---|---|---|---|---|---|---|---|
| 0.1 | 0.354 | 1.57e-39 | 9.20e-05 | | 0.1 | indep | causal | causal |
| 1.0 | - | 0.159 | 3.20e-49 | | 1.0 | - | indep | causal |
| 10 | - | - | 0.451 | | 10 | - | - | indep |

As for the experimental results of FCM3, we found that in the causal case, the confounding strength is a factor influencing the results more, which is different from the independent case. Moreover, we found that when the nonlinear function $f(X)$ is non-monotonic, such as $\sin(4X)$, the number of correct results is larger than the one in the linear case. Because the non-monotonic function itself can introduce a type of asymmetry which can indicate the causal direction for our method, the non-monotonic functions can be easier than the monotonic functions.

**Future work for dealing with the unknown confounders.** Causal discovery in the presence of unknown confounding is still an open problem, but there are some promising results in recent years such as (Janzing and Schölkopf, 2018; Mastakouri et al., 2021). Especially, (Mastakouri et al.,

Table 5: The experimental results of FCM3 (the causal case) with $f(X) = X$ in the presence of unknown confounding.

| wx\wy | 0.1 | 1.0 | 10 | 100 |
|---|---|---|---|---|
| 0.1 | 8.56e-56 | 2.98e-36 | 0.157 | 1.42e-06 |
| 1.0 | 1.22e-49 | 0.00027 | 9.60e-49 | 3.26e-51 |
| 10 | 0.002 | 7.02e-05 | 0.0018 | 0.007 |
| 100 | 0.385 | 0.398 | 0.577 | 0.70 |
| wx\wy | 0.1 | 1.0 | 10 | 100 |
| 0.1 | X cause | X cause | indep | X cause |
| 1.0 | X cause | X cause | Y cause | Y cause |
| 10 | Y cause | Y cause | Y cause | X cause |
| 100 | indep | indep | indep | indep |

Table 6: The experimental results of FCM3 (the causal case) with $f(X) = \sin(4X)$ in the presence of unknown confounding.

| $w_x$\$w_y$ | 0.1 | 1.0 | 10 | 100 |
|---|---|---|---|---|
| 0.1 | 1.42e-06 | 1.66e-68 | 5.62e-16 | 2.91e-09 |
| 1.0 | 5.68e-76 | 4.70e-81 | 5.62e-16 | 6.94e-51 |
| 10 | 2.17e-55 | 2.70e-47 | 1.36e-51 | 0.50 |
| 100 | 2.93e-06 | 2.70e-47 | 9.81e-24 | 0.08 |
| $w_x$\$w_y$ | 0.1 | 1.0 | 10 | 100 |
| 0.1 | X cause | X cause | X cause | X cause |
| 1.0 | X cause | X cause | X cause | Y cause |
| 10 | Y cause | Y cause | X cause | indep |
| 100 | Y cause | Y cause | X cause | indep |

2021) shows the identifiability results on time-series data in the presence of memoryless unknown confounders where the confounder doesn't have an autocorrelation effect. A potential research direction with our framework is that by recovering the corresponding dynamical process of a static causal discovery problem while considering the memoryless confounding, we can determine the causal direction between two variables in the presence of memoryless unknown confounders.

# D  PROOFS AND DERIVATION

## D.1  DERIVATION OF PROP. 1

Under constraints (I) and (II),

$$
\begin{aligned}
& W_2^2(p_0, p_T) \\
&= \int |M^*(\mathbf{x}_0) - \mathbf{x}_0|^2 p_0(\mathbf{x}_0) d\mathbf{x}_0 && \text{definition of } W_2^2 \\
&= \iint (f(E_x, E_y) - E_y)^2 p(E_y) dE_y p(E_x) dE_x && \text{Eqn. (6), and the independence of } E_x \text{ and } E_y \\
&= \mathbb{E}_{E_x}[W_2^2(p(E_y), p(Y|E_x))]
\end{aligned}
$$

## D.2  PROOF OF THE EXTENSION OF PROP. 1

Suppose that in the multivariate ANM, $\mathbf{x}_0 = [E_1, ..., E_m]'$, and $\mathbf{x}_T = [X_1, ..., X_m]'$, and $X_i = g_i(PA_i) + E_i$, where $PA_i$ denotes the parent of $X_i$.

$$W_2^2(p_0, p_T) = \int |M^*(\mathbf{x}_0) - \mathbf{x}_0|^2 p_0(\mathbf{x}_0) d\mathbf{x}_0$$

$$= \int |M^*([E_1, .., E_m]') - [E_1, .., E_m]'|^2 p(E_1, ..., E_m) d\mathbf{x}_0$$

$$= \sum_{i=1}^{m} \int |M_i^*([E_1, .., E_m]') - E_i|^2 p(E_1, ..., E_m) d\mathbf{x}_0,$$

(where $M_i^*$ represents the i-th element of $M^*$)

$$= \sum_{i=1}^{m} \int |M_i^*([E_1, .., E_m]') - E_i|^2 p(E_i) dE_i \, p(E_{1:m-i}) dE_{1:m-i},$$

(where $1 : m - i$ represents $1, ..., i - 1, i + 1, ..., m$,
and the independence of noise implies $p(E_i)p(E_{1:m-i})$)

$$= \sum_{i=1}^{m} \int |f(E_i, E_{1:m-i}) - E_i|^2 p(E_i) dE_i \, p(E_{1:m-i}) dE_{1:m-i}$$

$$= \sum_{i=1}^{m} \int |\widetilde{f}(E_i, M_{1:m-i}^*(E_{1:m-i})) - E_i|^2 p(E_i) dE_i \, p(E_{1:m-i}) dE_{1:m-i}$$

$$= \sum_{i=1}^{m} \int |\widetilde{f}(E_i, X_{1:m-i})) - E_i|^2 p(E_i) dE_i \, p(E_{1:m-i}) dE_{1:m-i},$$

(where applying the change of variable formula)

$$= \sum_{i=1}^{m} \int |\widetilde{f}(E_i, X_{1:m-i})) - E_i|^2 p(E_i) dE_i \, p(X_{1:m-i}) dX_{1:m-i} |det(J)|^{-1}$$

$$= \sum_{i=1}^{m} \int |\widetilde{f}(E_i, X_{1:m-i})) - E_i|^2 p(E_i) dE_i \, p(X_{1:m-i}) dX_{1:m-i},$$

(where $|det(J)|^{-1} = 1$ for ANMs)

$$= \sum_{i=1}^{m} \mathbb{E}_{PA_i}[\int |\widetilde{f}(E_i, PA_i)) - E_i|^2 p(E_i) dE_i]$$

$$= \sum_{i=1}^{m} \mathbb{E}_{PA_i}[W_2^2(p(E_i), p(X_i|PA_i))]$$

### D.3 PROOF OF THM. 1

**Part I.** We first derive the time interpolation $p(t, \mathbf{x}_t)$ of $p_0$ and $p_T$ under dynamical FCM constraints. Suppose that the FCMs are $Y = f(X, E_y)$. As for the dynamical formulation of the $L^2$ Wasserstein distance under constraints (I) and (II), according to the Jacobian equation, $p(t, \mathbf{x}_t) = p_0(\mathbf{x}_0)/|\det(J_{\frac{\partial \mathbf{x}_t}{\partial \mathbf{x}_0}})|$, $t \in (0, T)$, where $\mathbf{x}_t, \mathbf{x}_0 \in \mathbb{R}^2$ and $J_{\frac{\partial \mathbf{x}_t}{\partial \mathbf{x}_0}}$ is the Jacobian matrix w.r.t $\mathbf{x}_0$. Moreover, $\frac{\partial \log p(t, \mathbf{x}_t)}{\partial t} = -\frac{\frac{\partial}{\partial t}|\det(J_{\frac{\partial \mathbf{x}_t}{\partial \mathbf{x}_0}})|}{|\det(J_{\frac{\partial \mathbf{x}_t}{\partial \mathbf{x}_0}})|}$, where $J_{\frac{\partial \mathbf{x}_t}{\partial \mathbf{x}_0}} = \begin{bmatrix} 1 + \frac{t}{T}, & 0 \\ \frac{t}{T}\frac{\partial f}{\partial X}, & \frac{t}{T}(\frac{\partial f}{\partial E_y} - 1) + 1 \end{bmatrix}$.

According to Jacobi's formula and (Tao, 2013),

$$
\frac{d}{dt} |\det(A(t))|
$$

$$
= \pm \det(A(t)) \mathrm{tr}\left( A(t)^{-1} \frac{d}{dt} A(t) \right)
$$

$$
= \pm \det(A(t)) \mathrm{tr}\left( \det(A(t))^{-1} \mathrm{adj}(A(t)) \frac{d}{dt} A(t) \right)
$$

$$
= \pm \sum_{ij} \left( \mathrm{adj}(A(t))' \odot \frac{d}{dt} A(t) \right).
$$

where $\odot$ is element wise multiplication and $\mathrm{adj}(A)$ is adjugate matrix of $A$ and the sign takes the same sign as $\det(J_{\frac{\partial \mathbf{x}_t}{\partial \mathbf{x}_0}})$. Furthermore, since we know that under the structural constraints (I) and (II), $M^*$ is Eqn. (6). Thus, we replace $A(t)$ with $J_{\frac{\partial \mathbf{x}_t}{\partial \mathbf{x}_0}} = I + \frac{t}{T}(\nabla M^* - I)$, and then we have

$$
J_{\frac{\partial \mathbf{x}_t}{\partial \mathbf{x}_0}} = \left[ \begin{array}{cc} 1 + \frac{t}{T}, & 0 \\ \frac{t}{T}\frac{\partial f}{\partial X}, & \frac{t}{T}(\frac{\partial f}{\partial E_y} - 1) + 1 \end{array} \right];
$$

$$
\mathrm{adj}(J_{\frac{\partial \mathbf{x}_t}{\partial \mathbf{x}_0}})' = \left[ \begin{array}{cc} \frac{t}{T}(\frac{\partial f}{\partial E_y} - 1) + 1, & \frac{t}{T}\frac{\partial f}{\partial X} \\ 0, & 1 + \frac{t}{T} \end{array} \right];
$$

$$
\frac{d}{dt} J_{\frac{\partial \mathbf{x}_t}{\partial \mathbf{x}_0}} = \frac{1}{T}(\nabla M^* - I) = \left[ \begin{array}{cc} 0, & 0 \\ \frac{1}{T}\frac{\partial f}{\partial X}, & \frac{1}{T}(\frac{\partial f}{\partial E_y} - 1) \end{array} \right].
$$

Therefore,

$$
\frac{d}{dt} |\det(J_{\frac{\partial \mathbf{x}_t}{\partial \mathbf{x}_0}})| = \pm \frac{1}{T}(\frac{\partial f}{\partial E_y} - 1)(1 + \frac{t}{T}),
$$

which takes the same sign as $\det(J_{\frac{\partial \mathbf{x}_t}{\partial \mathbf{x}_0}})$.

**Part II.** We then proof the property of the corresponding dynamical systems of ANMs in Thm. 1.

Because $\frac{\partial \log p(t, \mathbf{x}_t)}{\partial t} = -\frac{\frac{\partial}{\partial t}|\det(J_{\frac{\partial \mathbf{x}_t}{\partial \mathbf{x}_0}})|}{|\det(J_{\frac{\partial \mathbf{x}_t}{\partial \mathbf{x}_0}})|}$, under the conditions in Thm. 1, $\frac{\partial f}{\partial E_y} = 1$ and it is obvious that $\frac{d}{dt}|\det(J_{\frac{\partial \mathbf{x}_t}{\partial \mathbf{x}_0}})| = 0$ and $|\det(J_{\frac{\partial \mathbf{x}_t}{\partial \mathbf{x}_0}})| \neq 0$, $\forall t \in [0, T]$. Furthermore, according to the theorem of instantaneous change of variables (Chen et al., 2018), which is a variant of Fokker-Plank equation, we know that

$$
\frac{\partial \log p(\mathbf{x}(t))}{\partial t} = -\mathrm{tr}(\nabla \mathbf{v}) = -\mathrm{div}\,\mathbf{v}.
$$

Therefore, $\mathrm{div}\,\mathbf{v} = 0$.

### D.4   PROOF OF PROP. 2

**Necessary direction.** We prove the necessary direction by showing that given $X$ is the direct cause of $Y$ in an ANM, $D(\mathbf{v}) = 0$. In the causal direction, because of the time evolution equation (5), we know that it is sufficient to check the divergence of the velocity field at time 0, and that the divergence taking zero value everywhere at time 0 implies the divergence taking zero value everywhere for $t \in [0, T]$. Because of Thm. 1, we know the $\mathrm{div}\,\mathbf{v} = 0$ at time 0. Therefore, $D(\mathbf{v}) = 0$.

**Sufficient direction.** We prove the sufficient direction by showing the contradiction with the identifiability of ANMs. Given that $D(\mathbf{v}) = 0$ under the constraints (I), (II), (III), and the identifiability conditions of ANMs, it tells us that there is an ANM in the form of Eqn. (1) which is consistent with the data distribution and has independent noise. Because of the identifiability of ANMs shown

by (Hoyer et al., 2008), there is no ANM with $Y \to X$ which is consistent with the data distribution and has independent noise at the same time. Therefore, it can only be the case the $X$ is the cause of $Y$.

Moreover, there is a concern that it may happen that in the direction which is not the causal direction, the model is not an ANM and its divergence measure is equal to zero. This may be problematic if the divergence measure is applied to applications. However, we will show that in general, this will not happen by proving that under weak assumptions $D(\mathbf{v}) = 0$ implies the model is an ANM. First, $D(\mathbf{v}) = 0$ implies div $\mathbf{v} = 0$ for all $\mathbf{x}_0$ with positive probability densities $p(\mathbf{x}_0) > 0$. In the following we say div $\mathbf{v} = 0$ in short. Second, div $\mathbf{v} = 0$ implies $\frac{\partial p(t, \mathbf{x}_t)}{\partial t} = 0$ a.e. according to Thm. 1. Consequently, $\frac{d |\det(J_{\frac{\partial \mathbf{x}_t}{\partial \mathbf{x}_0}})|}{dt} = 0$, which leads to $\frac{\partial f}{\partial E_y} = 1$ where $Y = f(X, E_y)$ a.e. according to the proof of Thm. 1. Therefore, under the assumptions: 1) $p(E_y)$ is positive in a continuous range of $E_y$; 2) in the range, $\frac{\partial f}{\partial E_y} = 1$ a.e. implies that it holds everywhere, we have $Y = E_y + C$, where $C$ is a quantity which doesn't change with $E_y$, e.g., it can be a function of $X$ or a constant. Therefore, $D(\mathbf{v}) = 0$ implies $Y = E_y + g(X)$ further under assumptions 1) and 2).

## E    IDENTIFIABILITY CONDITIONS OF ANMs IN (HOYER ET AL., 2008)

Because the identifiability conditions of ANMs in (Hoyer et al., 2008) are important and necessary for our Thm. 1, we include them here:

Let the joint probability density of $x$ and $y$ be given by $p(x, y) = p_n(y - f(x))p_x(x)$, where $p_n$, $p_x$ are probability densities on $\mathbb{R}$. If there is a backward model of the same form, i.e., $p(x, y) = p_{\tilde{n}}(x - g(y))p_y(y)$, then, denoting $\nu := \log p_n$ and $\xi := \log p_x$, the triple $(f, p_x, p_n)$ must satisfy the following differential equation for all $x, y$ with $\nu''(y - f(x))f'(x) \neq 0$:

$$\xi''' = \xi''(-\frac{\nu'''f'}{\nu''} + \frac{f''}{f'}) - 2\nu''f''f' + \nu'f''' + \frac{\nu'\nu'''f''f'}{\nu''} - \frac{\nu'(f'')^2}{f'},$$

where we have skipped the arguments $y - f(x)$, $x$, and $x$ for $\nu$, $\xi$, and $f$ and their derivatives, respectively. Moreover, if for a fixed pair $(f, \nu)$ there exists $y \in \mathbb{R}$ such that $\nu''(y - f(x))f'(x) \neq 0$ for all but a countable set of points $x \in \mathbb{R}$, the set of all $p_x$ for which $p$ has a backward model is contained in a 3-dimensional affine space.

## F    EXPERIMENTS AND DETAILS OF DIVOT

For the synthetic data experiments in Sec. 6, we visualized the generated data with the sample size 500 in Fig. 3. For determining causal direction, we used the variance-based divergence measure in

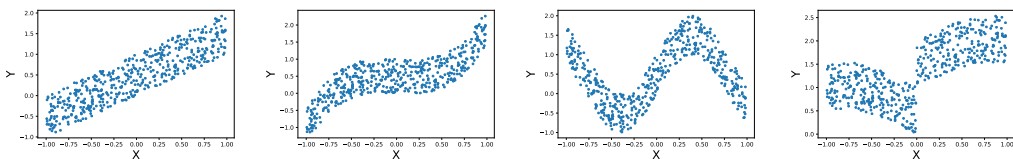

Figure 3: Synthetic data generated from ANMs with different (non)linear functions. From left to right, they are (1) $Y = X + E_y$; (2) $Y = 0.1(2.5X)^3 - 0.1X + E_y$; (3) $Y = \sin(4X) + E_y$; (4) $Y = 0.5X^3 - X + E_y$ if $x < 0$, and $Y = 1 - 0.5X^3 + X + E_y$, otherwise, where $X \sim \mathcal{U}(-1, 1)$ and $E_y \sim \mathcal{U}(0, 1)$ are uniform distributions.

Eqn. (11). The experiment setup is the same for experiments with different ANMs. For each ANM, we run experiments with different sample sizes, 10, 25, 50, 100, 200, and 500. For each sample size, 100 different datasets are generated. For all the synthetic data experiments in Sec. 6, we use batches for computing Eqn. (11) without using debiasing functions. In the following, we introduce positions (App. F.1), batches (App. F.2), debiasing functions (App. F.3), and optimization details (App. F.4) of DIVOT; as well as show its robustness to prior misspecification (App. F.5), more comparison with other benchmark methods (App. F.6), and its efficiency (App. F.7).

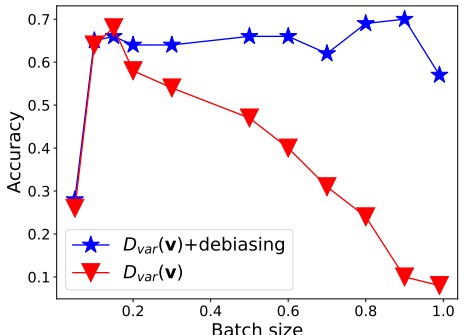 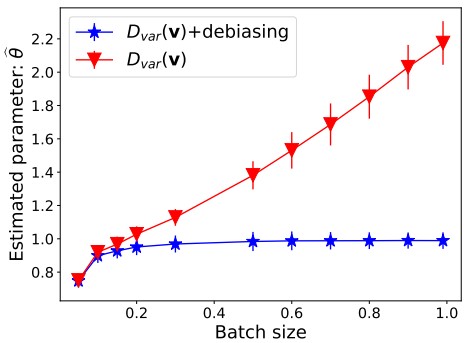

(a) The influence of batch sizes on determining causal direction and the improvement with a linear debiasing function.

(b) The influence of batch sizes on estimating $p(E_y; \theta)$ and the improvement with a linear debiasing function. The ground-truth $\theta$ is 1.

Figure 4: The influence of the batch size and the debiasing function on causal direction determination in panel (a) and density estimation in panel (b). The synthetic data (of which the sample size is 100) are generated from the linear ANM, $Y = X + E_y$, where $X \sim \mathcal{U}(-1, 1)$ and $E_y \sim \mathcal{U}(0, 1)$ are uniform distributions. In panels (a) and (b), the batch size is represented as the ratio of the sample size, e.g., $0.1$ represents that the batch size is $0.1 \times 100$; the red triangular represents the experiments with only using the variance-based divergence measure; the blue star represents using the the variance-based divergence measure and a linear debiasing function. The error bar in panel (b) represents standard deviation.

### F.1 NUMBER OF POSITIONS FOR COMPUTING THE VARIANCE-BASED DIVERGENCE MEASURE

Suppose a dataset $\{(x_i, y_i)\}_N$ with sample size $N$. For computing the divergence measure, we call $x \in \{x_i\}_N$ as a position and for each $x \in \{x_i\}_N$ we need to specify the corresponding vector $\overrightarrow{y_x}$, of which the number of elements is $N_x$ and generated samples of $\overrightarrow{e_y}$. In practice, it is not necessary to use all the $N$ positions for computing Eqn. (11). For the synthetic data experiments in Sec. 6, when $N > 50$, we choose 50 positions out of $N$; otherwise, we choose all the $N$ positions. To select a position (when $N > 50$), we first find the maximal and minimal values of $\{x_i\}_N$, compute the interval length, $\left(\max(\{x_i\}_N) - \min(\{x_i\}_N)\right) / 50$, and then choose a position every such length (or its nearest position available in the data). We can then compute Eqn. (11) for causal direction determination.

### F.2 BATCHES FOR THE FINITE (OR LIMITED) SAMPLE SCENARIO

In the finite sample scenario, it can be the case that there is no other data at a position $x$. Therefore, we use the neighbors of $x$ as a batch of data for computing Eqn. (11). We consider all the data in a batch as having the same value $x$. We represent the batch size with the percentage of the total number $N$ of samples. For the synthetic data experiments in Sec. 6, we use the batch size $0.4$ for the datasets with sample size 10; $0.2$ for the datasets with sample sizes 25 and 50; $0.15$ for the datasets with sample sizes 100 and 200; and $0.05$ for the datasets with sample size 500.

Note that for the synthetic data experiments in Sec. 6, the batch size is not larger than $0.2$ except the extremely small datasets with 10 samples. Because using larger batch sizes introduces bias by concatenating data at different positions in a batch. For the experiments in Fig. 4a, we generate a synthetic dataset with sample size 100 and use all the 100 positions for computing Eqn. (11). We can see that choosing a proper batch size can increase the accuracy of DIVOT by increasing the number of samples in a position; however, if further increasing the batch size such that it is larger than $0.2$, the accuracy is decreased because concatenating data at different positions leads to a different corresponding noise distribution compared with the ground-true $p(E_y)$. Thus, as shown in Fig. 4b, the estimation of the noise distribution becomes worse and worse with increasing the batch size. Such bias can be problematic especially in the few-sample case where we have to choose a larger batch size for computing the divergence measure. Therefore, we use a debiasing function such

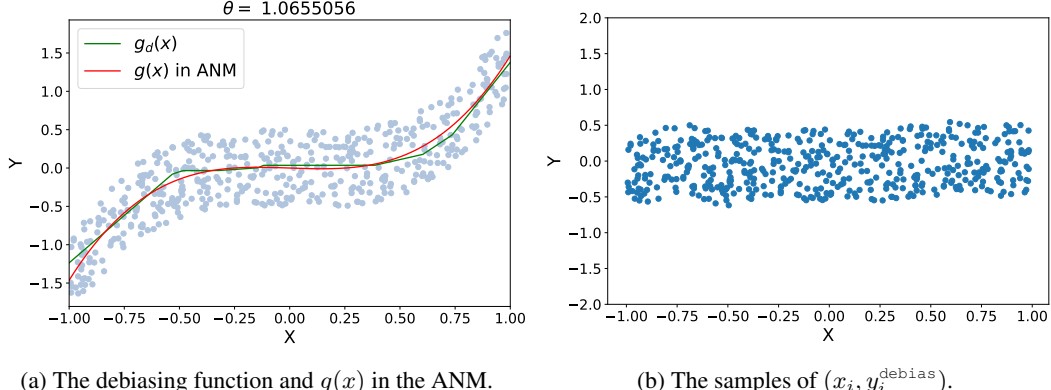

(a) The debiasing function and $g(x)$ in the ANM.

(b) The samples of $(x_i, y_i^{\texttt{debias}})$.

Figure 5: As a by-product, debiasing functions can be used for estimating the nonlinear function of an ANM in Eqn (1). We optimize the parameters of a neural network representing a nonlinear debiasing function and the parameter of a uniform distribution representing the hypothesized noise distribution. As for panel (a), the ground-true noise distribution is corresponding to $\theta = 1$ and the estimated parameter $\widehat{\theta}$ is 1.066. As for panel (b), given a dataset $\{(x_i, y_i)\}$, $y_i^{\texttt{debias}} = y_i - g_d(x_i)$.

that we can use a larger sample size for the few-sample case without sacrificing the performance of DIVOT.

## F.3 DEBIASING FUNCTIONS FOR THE FEW-SAMPLE SCENARIO

Because the mentioned bias is introduced by omitting the position information, we introduce it back with a debiasing function $g_d(x)$ to reduce the bias by modifying Eqn. (11) as

$$
D_{\text{var}}(\mathbf{v}) \approx \frac{1}{N} \sum_{x \in \{x_i\}_N} \frac{\left\| \text{sort}(\overrightarrow{y_x^{\texttt{debais}}}) - \text{sort}(\overrightarrow{e_y}) - \text{ave}(\overrightarrow{y_x^{\texttt{debais}}} - \overrightarrow{e_y}) \right\|_2^2}{N_x - 1}, \quad e_y^i \sim p(E_y; \theta),
$$
$$
\overrightarrow{y_x^{\texttt{debais}}} = \overrightarrow{y_x} - g_d(x; w), \tag{13}
$$

where $g_d(x; w)$ is a (non)linear function parameterized with $w$, e.g., in our experiments the linear debiasing function is $g_d(x) = w \times x$. As shown in Fig. 4, using the linear debiasing function guarantees not only the accuracy while using a larger batch size than 0.2, but also the correctness of the density estimation.

For a more complicated scenario, e.g., the ANM uses a nonlinear function, one can apply a neural network to the debiasing function. We found that as a by-product, a sufficient flexible debiasing function can be used for estimating $g(x)$ of ANMs in Eqn. (1). As shown in Fig. 5, we generate synthetic 500 data of an ANM with the nonlinear function $Y = g(X) + E_y = 0.1(2.5X)^3 - 0.1X + E_y$, where $X \sim \mathcal{U}(-1, 1)$ and $E_y \sim \mathcal{U}(-0.5, 0.5)$ satisfy uniform distribution. The $g_d(x)$ is close to the ground-truth $g(x)$. Because this is not the main focus of the work, we would like to refer the readers to the Jupyter notebook in supplementary materials for the details of the implementation.

Nevertheless, the purpose of using debiasing functions is not to estimate the $g(x)$ accurately but to reduce the bias introduced by using a large batch size. This means that when we use a restrictive class of $g_d(x)$, it can lead to a noticeable difference of $g_d(x)$ and $g(x)$. But for the purpose of causal direction determination, the divergence measure does not require to specify the nonlinear functional form of ANMs. Moreover, DIVOT based on optimal transport does not require to estimate $g(x)$ either. Therefore, DIVOT can still determine causal direction correctly in such case.

### F.4 OPTIMIZATION METHODS AND THE CONVEXITY OF THE OBJECTIVE FUNCTIONS OF DIVOT

In this section, we introduce the optimization method of the synthetic experiments in Sec. 6 and the convexity of the objective function. Next, we introduce the optimization method of DIVOT in the real-world data experiments in Sec. 6, of which the divergence measure uses a linear debiasing function and standard Gaussian noise.

As for the synthetic data experiments, we minimize the variance-based divergence measure in Eqn. (11) as an objective function, of which the parameter is $\theta$. Because when $\theta > 0$, the linear function $f_\theta^{\mathrm{noise}}(e_y^{\mathrm{source}})$ does not change the order of sorting results of $\overrightarrow{e_y^{\mathrm{source}}}$, we have

$$D_{\mathrm{var}}(\mathbf{v})$$

$$\approx \frac{1}{N} \sum_{x \in \{x_i\}_N} \frac{\left\| \mathrm{sort}(\overrightarrow{y_x}) - f_\theta^{\mathrm{noise}}(\mathrm{sort}(\overrightarrow{e_y^{\mathrm{source}}})) - \mathrm{ave}(\overrightarrow{y_x} - f_\theta^{\mathrm{noise}}(\overrightarrow{e_y^{\mathrm{source}}})) \right\|_2^2}{N_x - 1},$$

$$= \frac{1}{N} \sum_{x \in \{x_i\}_N} \frac{\left\| \mathrm{sort}(\overrightarrow{y_x}) - \theta \times \mathrm{sort}(\overrightarrow{e_y^{\mathrm{source}}}) - \mathrm{ave}(\overrightarrow{y_x} - \theta \times \overrightarrow{e_y^{\mathrm{source}}}) \right\|_2^2}{N_x - 1},$$

$$f_\theta^{\mathrm{noise}}(\overrightarrow{e_y^{\mathrm{source}}}) = \theta \times \overrightarrow{e_y^{\mathrm{source}}}, \quad \theta > 0, \quad e_y^{\mathrm{source}} \sim \mathcal{U}(0,1). \tag{14}$$

It is obvious that the divergence measure in Eqn. (14) is convex on $\theta > 0$. The convexity is achieved by parameterizing the noise distribution with a linear function $f_{\mathrm{noise}}$, and there are many exited toolboxes for solving the minimization problem.

For the synthetic data experiments, we used gradient descend for finding the optimal $\theta^*$. The gradient `jax.grad(loss)` is computed with the `autograd` in JAX (Bradbury et al., 2018). We update $\theta$ by specifying a step size `sz` and $\theta := \theta - $ `jax.grad(loss)` $\times$ `sz`. If after the update $\theta < 0$ (which has never happened), we set the value of $\theta$ as a positive number close to zero. We used `sz = 1` for all the synthetic data experiments. Moreover, because of the convexity, one can also use an one-step update method to directly get the best parameter $\theta^*$ by finding the root of the gradient function of Eqn. (14). We simply give a range of $\theta$ and find $\theta^*$ by a binary search method such that the gradient at $\theta^*$ is equal to zero. The experiments of DIVOT for ANMs in the real-world data experiments also used the one-step update for updating $\theta$ and the range $\theta$ is specified as $[0, 100]$. In addition, we need to optimize over the parameter $w$ of the linear debiasing function. We use gradient descent to find the best parameter, $w^*$. We update $\theta$ every 10 updates of $w$. For the PNL extension of DIVOT, we used the cyclic learning rate (Smith, 2017) with gradient descent. We update $\theta$ every 10 updates of $w$ and $\omega$ in (12). The program of DIVOT is terminated when the divergence measure converges. If a more complicated scenario requires $f_\theta^{\mathrm{noise}}$ to be a nonlinear function, one may need to use the gradient descend method instead of the one-step update. We find that it is sufficient to use the standard Gaussian distribution with the linear $f_\theta^{\mathrm{noise}}$ for DIVOT to have promising results in the experiments on the Tübingen datasets, which indicates that DIVOT is robust to the choice of models. We then investigate the robustness of DIVOT to prior misspecification.

### F.5 ROBUSTNESS TO PRIOR MISSPECIFICATION

We use different hypothesized noise distributions in DIVOT to test the robustness to the misspecification of noise distribution. The synthetic data are generated as in Sec. 6 with uniform distrbution noise $E_y \sim \mathcal{U}(0,1)$. In DIVOT, we use one of the three hypothesized distributions: uniform distribution $\mathcal{U}(0,1)$, beta distribution $\mathcal{B}(a = 0.5, b = 0.5)$, and standard normal distribution $\mathcal{N}(0,1)$. As shown in Fig. 6, DIVOT with the misspecified noise distributions has similar performance with the one using the correct class of distributions, which shows the robustness of DIVOT.

### F.6 COMPARISON WITH RESULTS OF BENCHMARK METHODS

We compare DIVOT with other benchmark methods, such as ANM (Hoyer et al., 2008), CAREFL (Khemakhem et al., 2021), RECI (Blöbaum et al., 2018), and LLR (Hyvärinen and Smith, 2013). As in (Khemakhem et al., 2021), the synthetic data are generated with the Laplace distribution as

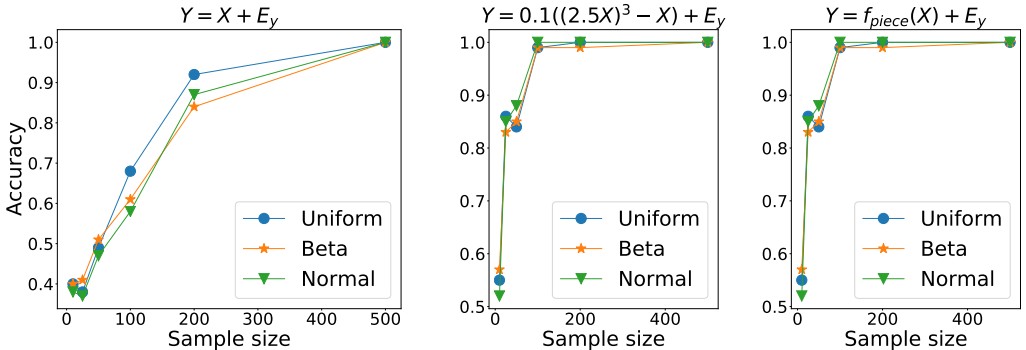

Figure 6: Robustness to prior misspecification: The performance of DIVOT with different hypothesized distributions of $p(E_y; \theta)$ on the datasets with uniform distribution noise $E_y \sim \mathcal{U}(0, 1)$.

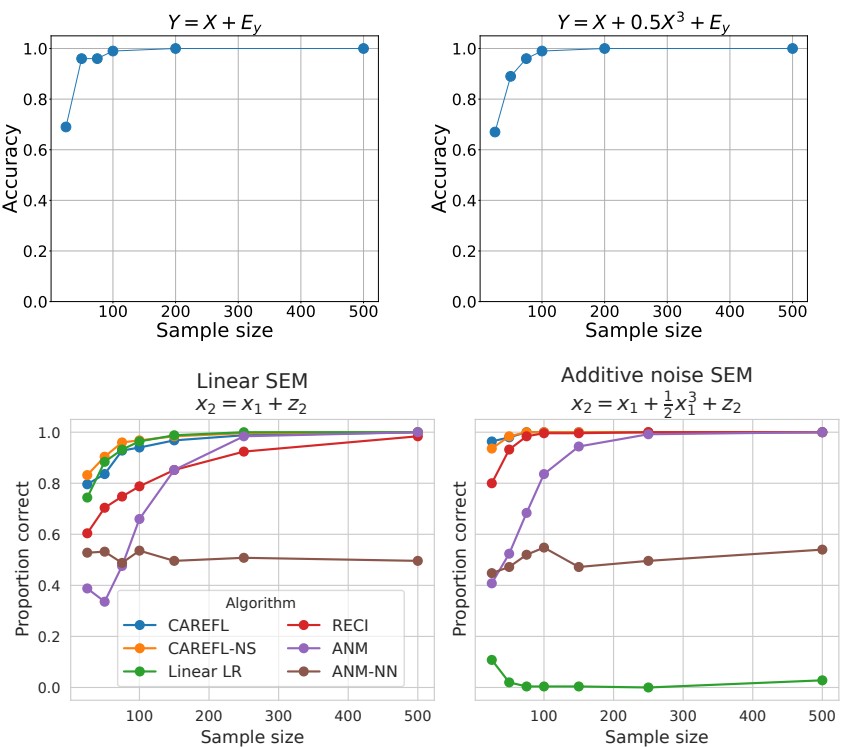

Figure 7: Comparison of the results of DIVOT (the first row) with the reported results in (Khemakhem et al., 2021) (the second row). The figures in the second row are taken from the screenshot of (Khemakhem et al., 2021). The noise ($Z_2$ and $E_y$) distribution of the synthetic data is a Laplace distribution. CAREFL represents the causal autoregressive model (Khemakhem et al., 2021); CAREFL-NS represents the causal autoregressive model without scaling; RECI represents the method, regression error causal inference (Blöbaum et al., 2018); ANM (Hoyer et al., 2008) uses Gaussian process while ANM-NN uses a neural network; linear LR is the linear likelihood ratio method (Hyvärinen and Smith, 2013).

the noise distribution. DIVOT uses batches without debiasing functions. And the data preprocessing is the same as in the real-world data experiments in Sec. 6. We run experiments on each dataset for 100 times. As shown in Fig. 7, for the sample size larger than 100 the accuracy of DIVOT is 100% which shows that DIVOT has promising results and performs better compared the other methods, especially in the linear case.

## F.7   EFFICIENCY OF DIVOT

For common causal discovery tasks in the bivariate case, the sample size $10000$ is a large and challengeable one. Causal discovery methods need to consider the efficiency of algorithms especially in the large sample size scenario. We apply DIVOT with batches and no debiasing functions to the synthetic data, of which the sample size is $10000$ generated with $Y = 0.1((2.5X)^3 - X) + E_y$ and $E_y \sim \mathcal{U}(0,1)$. As shown in Table 7, we test the running time of DIVOT to determine causal direction with different number of positions and different batch sizes. The experiments are based on MacBook Pro (15-inch, 2018) with 2.9 GHz 6-Core Intel Core i9. Our implementation is based on JAX (Bradbury et al., 2018) which uses Apache License and the running time is measured with the command `%timeit` in JAX.

Table 7: Running time of DIVOT on synthetic data, of which the sample size is $10000$. The DIVOT with batches and no debiasing function is tested with different number of positions and batch sizes.

| batch size | 50 positions | 100 positions |
|---|---|---|
| 0.001 | 240 ms ± 3.48 ms | 431 ms ± 4.33 ms |
| 0.01 | 7.71 s ± 475 ms | 14.3 s ± 262 ms |
| 0.1 | 1min 7s ± 1.81 s | 2min 19s ± 2.11 s |

