# OpenReview forum: "Optimal Transport for Causal Discovery"
_ICLR.cc/2022/Conference — ICLR 2022 Spotlight_

### Official Review · Reviewer_xS6T · 2021-10-29

**Correctness:** 4
**Technical Novelty And Significance:** 3
**Empirical Novelty And Significance:** 3
**Recommendation:** 6
**Confidence:** 4

**Main Review:**

As strenghts of the proposed method I see that is is nicely grounded on theorethical results, the paper is well structured, organized and written, and I personally enjoyed reading it.

Weaknesses, as weaknesses are cocerned I see the following main weak points:
1) it is not clear what happens, or at least I was not able to understand what happens in the case when the considered pair of variables are not related at all. Indeed, te proposed algorithm does not take into account this possibility. Lines 6-9 of Algorithm 1, which furthermore ignores that all we have is a sample and that which is which should be at least addressesd, what about the quantity you use are the same or nearly the same?
2) the decision to remove outliers is in itself something that practically could also be understood but from the theoretical point of view opens some doubts about the effectiveness of the proposed apporach. Indeed, standardization and subsequent removal of observations acconding to the 2 standard deviation limits brings bias in my opinion
3) numerical results seem to be in line with state of the art algorithms, however it seems that the proposed approach suffers from greater uncertainty about accuracy when compared to other literature approaches.

I also found some minor issues with typos.

**Summary Of The Paper:**

The paper tackles the problem of causal discovery in the basic case where a pair of variables is considered.
In particular, it is concerned with Functional Causal Models and how to establlish the causal direction for a pair of continuous variables.
It motivates the contibution by poiting out thet the performance of availabel algorithms is sensitive to the model assumptions, which
makes it difficult for practitioners to use them in real settings.
Then, the main contibution is a novel dynamical-system view of Functional Causal Models which aims to identify the causal direction in the case of pairs of variables. The paper exploits connection between functional causal models and optimal transport. Then, the paper studies the problem of optimal transport by taking into account the constraint originating from the considered functional causal model.

**Summary Of The Review:**

The paper tackles an interesting and well studied problem under a general setting, i.e. assuming uncounfondedness.
It is well written and proposes a nice method to discover the direction of causal relationship when considering pair of variables in isolation.
I have expected some more discussion about what to do in the case where the terms used in Algorithm 1 were not too different or extremely close. Furthermore, the results reported in Table 1 witness that the proposed methods achieves accuracy values similar to state of the art algorithms, while bringing more uncertainty about the true value of accuracy than other methods. Furthermore, for ANM the proposed method seems to be inferior to other state of the art algorithms.

---

> ### Author Response · Authors · 2021-11-22
> **Part 1a/5, the response to reviewer xS6T: The independent case**
>
> We would thank the reviewer for raising the point. To clarify the concern and provide more context in general, we will reply in the following five parts:
>
> 1. we show that this is an important point and we follow the convention of bivariate causal discovery literature;
> 2. we introduce the reason for such convention;
> 3. we show how to modify the output conditions to cover the independent case based on our theoretical basis in the paper;
> 4. we also introduce how to use bootstrapping for providing the significance of our results;
> 5. additional experiments are provided for the modification.
>
> _We **have added** the modification with the experimental results **in the App. A** of the paper._

---

> > ### Author Response · Authors · 2021-11-22
> > **Part 1b/5, the response to reviewer xS6T: The independent case**
> >
> > **1) The convention.** It might indeed be perceived as strange why works about bivariate causal discovery, such as (Monti et al., 2020), (Zhang and Hyvärinen, 2009b), (Blöbaum et al., 2018), (Khemakhem et al., 2021), (Compton et al., 2021), and others, rarely discuss the case where the two variables are not related at all. Currently, our work also follows this convention, but in the following, we will show the reason for such a convention and show that our method with a minor modification on the output condition is able to deal with the independent case. The modification together with the other discussion and extension is a direct extension based on the provided theoretical foundation in the paper.
> >
> > **2) Reason for convention.**
> > **_First_**, in general, the independent case is less problematic (or even not a problem) than the correlated case. With the existing methods, we can deal with the independence case, such as using independence tests or constraint-based methods; however, in the correlated case, without introducing further assumptions, the two causal-direction candidates are in the Markov equivalent class and the causal direction is not identifiable.
> > **_Second_**, as in (Zhang and Hyvärinen, 2009b) and (Monti et al., 2020), they apply the bivariate causal discovery methods to the results of score/constraint-based methods (such as PC or GES) in order to deal with the multivariate case and further determine the undirected edges. (If the reviewer is interested in this point, we would like to refer to the discussion about **_our multivariate extension_** in the response to **_reviewer GA77_**). Therefore, the essential point of a bivariate causal discovery method is to determine whether X causes Y or Y causes X.
> >
> > **3) Modification of the output conditions.**
> > If one wants to include the independent case in the algorithm, our proposed measure is able to deal with the independent case without relying on other methods or tests after adapting the output conditions of the algorithm. The values of our proposed measure in the independent case are zero in the two directions, while the measure values of the correlated case are a zero value in one direction and a non-zero value in another direction. The modified output conditions are shown in the table.
> >
> > | Output of DIVOT | Div(X->Y) = 0 | Div(X<-Y) = 0 |
> > |-----------------|---------------|---------------|
> > | X indep Y       | True          | True          |
> > | X -> Y          | True          | False         |
> > | X <- Y          | False         | True          |
> >
> > To determine whether the value of our measure is zero or not, one could follow a similar way as (Blöbaum et al., 2018) choosing a threshold for real-world applications practically. However, different applications may have different thresholds. For example, in practice, when the noise distributions have different variance in the finite/few sample scenario, the larger variance can lead to the larger measure value. Although one can handle the problem well by normalizing the variance-based measure value with the estimated noise variance,  deriving a statistical test for the finite/few sample case is still the ideal way, which is another nontrivial task without assuming the type of noise distribution and will be the future work of our method.
> >
> > **4) Bootstrapping for the significance of the results.** Therefore, instead of testing whether a measure value is significantly zero or non-zero, we suggest using a bootstrapping method and testing whether the two measure values are significantly different (one could also test whether the difference between the two measure values are significantly zero or not):
> > Step 1. use bootstrapping (resampling with replacement) to get B bootstrapping datasets;
> > Step 2. compute two measure values in the two directions for each bootstrapping dataset;
> > Step 3. apply a two-sample test (T-test) to check whether the mean of one measure value in one direction is significantly different from another one in the other direction.
> > Step 4. if two measure values are significantly different, we then conclude that the smaller one is in the causal direction; if they are not significantly different, we then conclude that it is the independent case.
> >
> > **5) Additional experimental results.**
> > We also did some experiments for showing the effectiveness of our modified algorithm. Firstly, we generate five synthetic datasets with 1000 samples for each ANM in the synthetic data experiments in Sec. 6 and a pair of independent variables. For each dataset, we generate 50 bootstrapping datasets. The p values and the results with the significant level at 0.05 are shown in the following table.
> >
> > |         | M1        | M2        | M3        | M4        | independent case |
> > |---------|-----------|-----------|-----------|-----------|------------------|
> > | p value | 1.126e-54 | 3.917e-65 | 7.855e-64 | 1.087e-51 | 0.2555           |
> > | result  | X cause   | X cause   | X cause   | X cause   | independent      |

---

> > > ### Author Response · Authors · 2021-11-22
> > > **Part 2/5, the response to reviewer xS6T: The extremely close case**
> > >
> > > We would like to thank the reviewer for pointing this out. Roughly speaking, the extreme case (where the measure values in the two directions are close) can happen when the causal relationship is so weak that the FCM is similar to the independent case (e.g., when the coefficient of X is extremely small in the linear case, it is close to the independent case where the measure values are close). If this is the case, we would suggest using the bootstrapping method, which can in practice well reflect the confidence/significance of results. We did some experiments regarding the reviewer's concern. We generate four datasets with 1000 samples for the four FCMs in the synthetic data experiments in Sec. 6 with a weighting factor w:  Y = w * f(X) + Ey, and then use the bootstrapping method and T-test. The table shows the p values, of which small values indicate the significantly different measure values. If the significant level is 0.05, our method can tell the difference between two measures in the causal case when w > 0.03. Moreover, it can also happen in the presence of unknown confounders, we would also like to refer the reviewer to our response to **_reviewer pbTi about unknown confounding_**.
> > >
> > > | w  | 0.01  | 0.02  | 0.03   | 0.04      | 0.05      |
> > > |----|-------|-------|--------|-----------|-----------|
> > > | M1 | 0.155 | 0.038 | 0.0006 | 2.429e-08 | 3.091e-11 |
> > > | M2 | 0.251 | 0.070 | 0.0003 | 0.0001    | 1.935e-07 |
> > > | M3 | 0.390 | 0.119 | 0.0196 | 0.0003    | 9.578e-08 |
> > > | M4 | 0.408 | 0.098 | 0.0037 | 2.571e-05 | 2.346e-11 |
> > >
> > > | w  | 0.01  | 0.02    | 0.03    | 0.04    | 0.05    |
> > > |----|-------|---------|---------|---------|---------|
> > > | M1 | indep | X cause | X cause | X cause | X cause |
> > > | M2 | indep | indep   | X cause | X cause | X cause |
> > > | M3 | indep | indep   | X cause | X cause | X cause |
> > > | M4 | indep | indep   | X cause | X cause | X cause |

---

> > > > ### Author Response · Authors · 2021-11-22
> > > > **Part 3/5, the response to reviewer xS6T: The outlier issue**
> > > >
> > > > As for our experiments on the Tübingen cause-effect datasets, we are not doing some specific pre-processing to bias the performance. And it is also fine with other techniques to deal with outliers. If the reviewer is still doubting it, one could also use resampling/bootstrapping for getting B bootstrapping datasets (e.g., B = 50) and then make the final decision based on the B bootstrapped datasets. Moreover, regarding the effectiveness of our method, the reviewer could also consider the experimental results on the synthetic datasets which follows the same procedures as (Khemakhem et al., 2021), and our newly provided experimental results with bootstrapping datasets in the response to **_the independent case and the extreme case_** discussing the significance of a result and the performance in the presence of **_unknown confounding in the response to reviewer pbTi_**.
> > > >
> > > > Because our proposed method is based on optimal transport, technically, on the one hand, it has many beneficial properties such as the efficiency of computing the measure without learning a regression model and robustness to the noise misspecification; on other hand, it is also facing the same challenges as optimal transport, e.g., the outlier issue. Especially, our method relies on the couplings of the generated noise samples and the observed data points given by the L2 Wasserstein distance. Nevertheless, developing outlier-robust optimal transport is an active, ongoing, and popular research topic (A).  With the development of the optimal transport methods, our method will also have a more promising development and result.
> > > >
> > > > (A) Mukherjee, Debarghya, Aritra Guha, Justin M. Solomon, Yuekai Sun, and Mikhail Yurochkin. "Outlier-robust optimal transport." In International Conference on Machine Learning, pp. 7850-7860. PMLR, 2021.

---

> > > > > ### Author Response · Authors · 2021-11-22
> > > > > **Part 4/5, the response to reviewer xS6T: Reviewer's comment: Bivariate causal discovery is a well-studied problem under the causal sufficiency assumption.**
> > > > >
> > > > > We sincerely disagree with the reviewer xS6T that it is a well-studied problem under the causal sufficiency assumption, instead, it is an active research direction, e.g., (Monti et al., 2020), (B), (C), (Blöbaum et al., 2018), (Khemakhem et al., 2021), and (Compton et al., 2021). As what _reviewer pbTi_ said, what we are working on is a fundamental and challenging task. The field, causal discovery, indeed achieved much development and improvement, e.g., the identifiability results for FCMs, but it is far from ideal. Many works about it cannot imply how well it is studied; on contrary, it indicates how difficult is the problem. One may argue that identifiable additive noise models or post-nonlinear models are sufficient. But the field causal discovery is different from a domain application, instead it is a general problem across multiple disciplines of science. Therefore, it requires further extending the existed FCMs to be more general to fit the requirements. Unfortunately, it is nontrivial to either understand the technical model assumptions or extend the functional classes. Hence, our work first comes in to understand the FCMs and the constraints with dynamical systems, and proposes a criterion based on the new perspective. Hopefully, we could further extend the existed identifiable FCMs with the help of dynamical systems in the future.
> > > > >
> > > > > (B) Janzing, Dominik. "Statistical Asymmetries Between Cause and Effect." In Time in Physics, pp. 129-139. Birkhäuser, Cham, 2017.
> > > > >
> > > > > (C) Janzing, Dominik. "The cause-effect problem: Motivation, ideas, and popular misconceptions." In Cause Effect Pairs in Machine Learning, pp. 3-26. Springer, Cham, 2019.

---

> > > > > > ### Author Response · Authors · 2021-11-22
> > > > > > **Part 5/5, the response to reviewer xS6T: Experiments on Tübingen benchmark datasets**
> > > > > >
> > > > > > Regarding our experiments, because causal discovery is not a specific domain application, which brings more difficulties to create benchmark datasets and benchmark the performance of methods, we choose the (only) suitable one among the few available benchmark datasets for causal discovery. Tübingen cause-effect dataset as a benchmark dataset consists of different cause-effect pair datasets in different applications. Note that the causal direction of one cause-effect pair is identifiable doesn't mean that the others in the datasets are also identifiable, e.g., one can check the scatter plots of the datasets and find that there are several groups with similar plots and that some groups are clearly linear Gaussian. Therefore, our performance on such datasets actually reflects the uncertainty and the non-identifiability fact of some groups of cause-effect pairs in the benchmark dataset. In other words, given the current identifiability results of causal discovery, after a certain level, the performance on such datasets is expected to be confidently wrong or uncertain about the results. For example, the standard deviation of all experiments in IGCI (Janzing et al., 2012) is 7 percent based on 70 pairs; and the standard deviation of (Compton et al., 2021) is 10 percent.

---

> > > > > > ### Comment · Reviewer_xS6T · 2021-11-22
> > > > > > **I wrote well studied and not that it is a closed problem**
> > > > > >
> > > > > > Just a first objection to your disagreement, I wrote that the problem is well studied and not that it is simple nor a closed problem.
> > > > > > Apologize if the well studied was intended as the author/s was/were studing a minor problem.
> > > > > > Language is often ambiguos and serving as a reviewer, as well as you also know, asks for many efforts.

---

> > ### Comment · Reviewer_xS6T · 2021-11-23
> > **I appreciated your rebuttal on all the issues from my review**
> >
> > Dear author/s,
> > I greatly appreciated you rebuttal, which helped me to better appreciate your work.
> > Therefore, I'll improve my rating of your work thanks to your answers to questions from other reviewers. Indeed, in the light of the new release of your paper my understanding of your work was significantly improved.

---

> > > ### Author Response · Authors · 2021-11-23
> > > **We thank the reviewer for the encouraging reply**
> > >
> > > Thank you so much for your time and review! It has been very valuable and helpful for us to further improve our work. Your encouraging reply is appreciated very much by us. We are so grateful for and looking forward to the increased score.

---

### Official Review · Reviewer_pbTi · 2021-11-03

**Correctness:** 4
**Technical Novelty And Significance:** 4
**Empirical Novelty And Significance:** 3
**Recommendation:** 8
**Confidence:** 4

**Main Review:**

Good and very interesting paper on a well-known but challenging problem. In particular the derivation of the new measure through the connection with dynamical systems is both intriguing and inspiring.

Perhaps main drawback is that the information density in the paper is quite high and technical, and assumes familiarity of the reader with literature from other fields that may not be well-known to the wider ICLR audience, making it hard to understand certain parts of the paper in isolation. Even with a background in physics I had think back hard to remember details of vector field divergences and incompressible flow mechanics. That said, the ideas in the paper are well motivated and supported by rigorous theoretical derivations, making it sufficiently worthwhile for an interested reader to spend the extra effort.
I briefly entertained the hope the approach might also apply in the presence of confounding, but as with similar approaches it does still require causal sufficiency and additive noise components. Performance shows improvement on existing methods, though still in the somewhat disappointing range of around 75% accuracy for a collection of 50-50 decisions. But that just shows the fundamental difficulty of the problem.

In short: good paper on challenging subject, bit technical but clear accept.

Few remarks/questions:
p1, bottom: ‘.. of which the modern form was initiated by ..’, ‘.. by augmenting a time-dimension …’, ‘ … in the context of the Jacobian problem ..’ => these references are unlikely to be known by the majority of readers, and it would help to give just a minor hint of what these might entail
p2,top, point 2. ‘pressureless flow’ => do you mean the more standard notion of ‘incompressible flow’? or else explain the difference
p2,mid: ‘.. from p_0(x) to p_T(x) …’ => technically the random variables do not need to obtain their value after a constant time T, e.g. different coin throws end up heads or tails, but in the actual underlying dynamical system different throws can take different times to come to rest. Does this impact the interpretation in terms of the map M or is the actual value of T essentially irrelevant? If so then explain or perhaps remove it form the description.
p3,bot: ‘The sense of causality is enforced .. FCMs’ => this sentence is overly vague / ambiguous: please clarify
p4, final sentence above 3.2 .’Therefore optimality .. problem.’ => the implication of this sentence is unclear
p5, Prop2: can the authors comment/speculate on whether in the presence of unobserved confounders, in almost all cases we would expect to find D(v) =/= 0 in both the causal and anti-causal direction ?
idem: ‘ .. and the identifiability conditions of ANMs (Hoyer,2008) …’ => these conditions are crucial and should be explicitly stated in the current paper
p7, Ext. to PNL: it is not entirely clear how this invertible function should be chosen based on a given data set
p7, ‘Minimisation’ , ‘Note .. is not necessary to be the true distribution or even significantly different from the true one’ => this sentence is unclear


**Summary Of The Paper:**

The paper considers the problem of establishing causal direction in a bivariate system under the assumptions of additive noise (possibly post-nonlinear) and no unobserved confounding. It establishes a connection between functional causal models (FCM) and optimal transport problems in dynamical systems to derive a new divergence metric that can be efficiently computed as a conditional variance estimate, and shows this quantity is zero iff in the causal direction..Efficacy of the metric is evaluated on synthetic data as well as the real-world Tuebingen data stand found to outperform existing measures.

**Summary Of The Review:**

Good and inspiring paper on challenging subject, bit technical but clear accept.

---

> ### Author Response · Authors · 2021-11-22
> **Part 1/2, the response to reviewer pbTi: The scenario in the presence of unknown confounding**
>
> We would thank the reviewer for the time and feedback on our work. We are so grateful for the encouraging comments, the elegant summary, helpful suggestions, and the opinion on the Tübingen benchmark datasets.
>
> **Unknown confounding.** We would thank the reviewer for raising the point. To answer the reviewer's question, we have **added** the detailed **analysis** and the **discussion** together with additional **experimental results** for the scenario in the presence of **unknown confounding in App. C of the updated paper**. In summary, the unknown confounding has different influences on the results of our method in different situations: 1) **_independent case:_** when two variables are independent; 2) **_causal case:_** when there is a causal relationship between two variables. Depending on how the unknown confounding influences the pair of variables, it can make our method
>
> 1. reverse the direction of the result in the causal case;
>
> 2. disregard the causal relationship in the causal case;
>
> 3. introduce extraneous causal relationship in the independent case.

---

> > ### Author Response · Authors · 2021-11-22
> > **Part 2/2, the response to reviewer pbTi: Remarks and questions**
> >
> > **Incompressible flow and pressureless flow.**
> > We would thank the reviewer for asking the question about the two flows. _We have **updated** the paper respectively **after Eqn. (6)** and highlighted with the blue text._ It is indeed a confusing concept which is not well defined in (Benamou and Brenier, 2000). We have tried to interpret it with free particles after Eqn. (6) before Eqn. (7) in the paper, but we will further explain it. To clarify it, a pressureless flow is not necessary an incompressible flow, and vice versa. As mentioned by (Benamou and Brenier, 2000), the pressureless flow is a "crude" model, which is as its name meaning that there is no pressure in the system and that fluid particles are not subject to any pressure or force. Therefore, given the initial positions and velocities or given the initial and final positions, the trajectories are determined. In contrast, incompressible flows are defined with the zero divergence of its velocity field. _Interestingly, the corresponding flows of ANMs are pressureless as well as incompressible._ Then can we use a more general flow for PNLs or the other general FCMs? This is actually directly indicating another potential direction for future works, which generalize the dynamical systems (pressureless flows) to more flexible systems such as (in)compressible flows. There are also many studies about the (in)compressible flows and optimal transport:
> >
> > Brenier, Yann. "The least action principle and the related concept of generalized flows for incompressible perfect fluids." Journal of the American Mathematical Society 2, no. 2 (1989): 225-255.
> >
> > Brenier, Yann. "Connections between optimal transport, combinatorial optimization and hydrodynamics." ESAIM: Mathematical Modelling and Numerical Analysis 49, no. 6 (2015): 1593-1605.
> >
> > Liu, Jian-Guo, Robert L. Pego, and Dejan Slepcev. "Least action principles for incompressible flows and optimal transport between shapes." arXiv preprint arXiv:1604.03387 (2016).
> >
> > **Invertible function in PNL.** We agree that the invertible function in Eqn. (14) here is too simple to fit the general purpose of applications. For simplicity, we introduced Eqn. (14) as a demonstration/an example for constructing an invertible function and extending to the PNL case, rather than proposing a flexible PNL model for applications. Therefore, we would suggest using a more flexible model depending on applications, e.g., as in (Zhang et al., 2015a), using more tanh(.) modules in warped Gaussian process, and the invertible neural networks used in normalizing flows.
> >
> > **p2,mid: ... from p0(x) to pT(x)...** As what reviewer said, the time T is irrelevant, which is used for representing another density function different from $p_0(x)$. We were trying to modify the notation of (Benamou and Brenier, 2000) a bit to make it consistent with our notation. In fact, (Benamou and Brenier, 2000) uses density functions $\rho_0(x)$ and $\rho_T(x)$ without the notion of probability. Therefore, thank the reviewer so much for the question. _We have **updated** the paper **around Eqn. (1)** and highlighted with the blue text with the original notation for introducing the context._
> >
> > Regarding the meaning of T, for the static problem, we have no access to the time information, e.g., in bivariate causal discovery setting, the observed variables (X,Y) are represented by $\mathbf{x}_T$ with its probability density function $p(X,Y)$ denoted by $p_T(\mathbf{x})$, and their noise terms ($E_x$,$E_y$) in FCMs are represented by $\mathbf{x}_0$ with its probability density function $p(E_x,E_y)$ denoted by $p_0(\mathbf{x})$. $T$ comes in when augmenting a time dimension and considering $p(E_x,E_y)$ as the initial-time probability density function and $p(X,Y)$ as the final-time probability density function. It is not the real-time of the underlying process which may take different time to collect the data of $X$ and $Y$, instead the time for construction.
> >
> > **Minimization.**
> > In the method DIVOT, it requires specifying a distribution for the noise $E_y$, such as normal distribution N(0,2); however, N(0,2) may not be the distribution of $E_y$ in the ground-truth FCM which is used for generating the data. In other words, the distribution of $E_y$ initialized with an arbitrary parameter $\theta$ may not be the true distribution of $E_y$. So we need to optimize over the parameter $\theta$.
> >
> > **Clarification.**
> > We would thank the reviewer for pointing out the unclear parts and helping us improve the paper:
> > "p1, bottom: minor hint; identifiability conditions of ANMs; p3, bottom: the sense of causality is enforced .. FCMs; p4, final sentence above 3.2."
> > _We have updated the paper respectively **in Sec. 1, App. F, the bottom of p3, and above Sec. 3.2** with the blue text in the updated paper._

---

### Official Review · Reviewer_GA77 · 2021-11-04

**Correctness:** 4
**Technical Novelty And Significance:** 4
**Empirical Novelty And Significance:** 3
**Recommendation:** 6
**Confidence:** 3

**Main Review:**

I believe that the authors' attempt of reformulating bivariate causal discovery problem is a novel and inspiring endeavor. Given that the problem of causal discovery is a conceptual problem as much as it is a statistical problem, the introduction of such new perspectives are certainly helpful for the discussion in this field. The authors' presentation is concise and clear, and the paper is easy to follow (modulo the additional comments I provide below).

One potential avenue for improvement is the motivation of the present reframing. What shortcoming of the information geometric, information theoretic, probabilistic, or more recently, topological analyses of the problem of causal discovery motivated the authors to present an optimal transport-based perspective? What additional advantages do the authors' formulation (or its potential sequels) provide to our understanding and methodology regarding causality? I find their work interesting regardless, but the preceding point is crucial for any work that formulates a novel mathematical framework causal discovery.

The authors limit their presentation to bivariate causal discovery problem. I do not think this is necessarily a crippling limitation, given the novelty of their theoretical perspective. However, the authors still need to provide an outlook for other cases as the authors do not seem to position their work as a purely conceptual development. If practical implications are to be considered, at least initial discussions of the multivariate case, as well as other standard challenges such as latent confounding, selection bias etc. should be provided.

Another potential development of the current work is to improve its presentation. If accepted, I encourage the authors to use the extra space conferred by the camera-ready version to provide a better introduction to the field of optimal transport, its prevalent use cases, as well as various concepts that are used in the field. The authors could also consider moving the related work section to the end of the introduction to further strengthen their initial setup.

**Summary Of The Paper:**

The authors frame the bivariate causal discovery problem in terms of the analysis of a dynamical-system. They use results from the field of optimal transport to interpret additive noise models from this framework. They also develop a novel criterion and a causal discovery algorithm based thereupon, and compare their results with previous methods used in bivariate causal discovery.

**Summary Of The Review:**

The current paper proposes a novel and well-presented formulation of the causal discovery problem, and thus is of interest to the conference audience. Improved comparison with existing frameworks and expanded discussion regarding practical use cases are recommended to improve the paper.

---

> ### Author Response · Authors · 2021-11-22
> **Part 1/3, the response to reviewer GA77: Motivation, contribution, and related works**
>
> We would thank the reviewer for providing the constructive feedback. However, there seems to be a misunderstanding about the motivation and contributions of our work. The reviewer seems to consider our contributions as purely proposing a practical measure for bivariate causal discovery. In fact, we do position our work as a conceptual development which provides a new perspective of FCMs, and the proposed measure (DIVOT) is one application of utilizing such new perspective for causal discovery. Our contributions are actually **two-fold**: **_1) conceptual-level; 2) technical-level_**. We will elaborate them in the following.
>
> **The motivation and the ultimate goal** of our work is to generalize the identifiable FCMs and relax the restrictive assumptions, which is nontrivial, not clear how to achieve, and in huge demand.
> **Our first-fold contribution** is conceptual-level. We figure out a way to connect the two fields, dynamical systems, and FCM-based causal discovery, and prepare the theoretical foundation for the future work/next step. Hopefully, by leveraging the connection of dynamical systems and causal discovery, identifiable FCMs can be further generalized with the richer functional classes given by the dynamical models. More specifically, the connection gives us a new perspective of FCMs. With the new view, we can specify the corresponding dynamical systems of FCMs and study the properties of such dynamical systems. We then try to utilize the new perspective and provide more insights for FCMs and the assumptions of FCMs; hence, as an example we thoroughly introduced the corresponding dynamical systems of ANMs,  which directly leads to a new bivariate causal discovery criterion based on the dynamical system view.
> **The second-fold contribution** is technical-level. We study the (dynamical) optimal transport under FCM constraints and provide an efficient way to solve it. It is still valid in the multivariate case to solve high-dimensional optimal transport problems under FCM constraints as shown **_in the response to the multivariate extension_**. Moreover, we provide a new criterion, define a valid measure for it, and further propose a simple variance-based implementation which inherits many benefits from optimal transport.
>
> Regarding what the reviewer mentioned about the **related work**, we are grateful for the comments on it, and we plan to add more comparisons in the final version of the paper. (If the space is not enough, we would include more comparison in the appendix). **_The reason_** why we put the related work section after the method section is because of the **_two-fold contribution structure_**. Our first-fold conceptual-level contribution is unique and novel, which is the essential core part of the paper. We considered it as a valuable and important contribution for the further development of FCMs in the field. So we spent much effort and space in the first three sections to elaborate the connection and provide insights for the new perspective of FCMs. An additional reason for putting the related work section after the proposed method and before the experiment section is to provide **_a better context_** for the comparison to other methods in the experiment section.

---

> > ### Author Response · Authors · 2021-11-22
> > **Part 2/3, the response to reviewer GA77: Discussion about the unknown confounding and the multi-variate case**
> >
> > We would thank the reviewer for the suggestions. We limit ourselves to the two-variable case for the sake of clarity and simplicity. Indeed, our method can be directly extended to the multivariate case. Extending our work to the multivariate case, in fact, on its own can be another score-based causal discovery method, and has its own problems and technical details which are irrelevant and out of the scope of this work. But we agree with the reviewer that it would be helpful and beneficial for the readers to mention practical implications in the paper, and we have added the outlook of the multivariate case in the paper as well as the discussion and experimental results in the presence of unknown confounding.
> >
> > _**As shown in the updated paper**, we have added the **extension to the multivariate case** in the end of **Sec. 4, App. B, and App. D.2** with the blue text and the margin comment "NEW (GA77)"._
> > A short summary is that:
> >
> > 1. Similar to the other bivariate methods, our method can be applied in a naive way to the multivariate case using skeleton search methods first;
> >
> > 2. We show the limitation of the previous naive methods and propose a new extension of our method for the multivariate case where our measure can be applied directly. In particular,
> >
> > 	a) we summarize the good properties of our extended measure;
> >
> > 	b) we extended Prop. 1 and variance-based divergence measure to the multivariate case;
> >
> > 	c) we provided the direct extension for orienting the edges of a given causal skeleton, and introduced several points in the multivariate case for future works to further explore.
> >
> > Regarding the discussion and the additional experimental results for **the unknown confounding case**, we would like to refer the reviewer to **_our response to reviewer pbTi_**.

---

> > > ### Author Response · Authors · 2021-11-22
> > > **Part 3/3, the response to reviewer GA77: Introduction of optimal transport**
> > >
> > > We are grateful for the reviewer's suggestion. **_We have updated our introduction section and highlighted it with blue text._** We will add more introduction about optimal transport in the appendix.

---

> > > > ### Comment · Reviewer_GA77 · 2021-11-29
> > > > **Thank you for the response**
> > > >
> > > > I thank the authors for their detailed response. I did not assume that the authors' contribution was limited to the bivariate case, actually that was the reason I thought extra clarification would be helpful, which the authors provided.
> > > >
> > > > I think the conference audience would benefit from the current work, and therefore recommend its acceptance.

---

> > > > > ### Author Response · Authors · 2021-11-30
> > > > > **Your time and review are appreciated very much.**
> > > > >
> > > > > We would thank the reviewer so much for the time, the review, and the suggestion. We are grateful for the comments and feedback. They helped us a lot to further improve the work.

---

### Decision · Program_Chairs · 2022-01-20

**Decision:**

Accept (Spotlight)

**Comment:**

This paper is a solid contribution to researchers in this field, as it provides a new idea for the basic problem of determining the direction of causality between two variables, using the functional causal model as a dynamical system and optimal transport.